# Determinants of sugar-induced influx in the mammalian fructose transporter GLUT5

**Sarah E McComas[1], Tom Reichenbach[1], Darko Mitrovic[2], Claudia Alleva[1], Marta Bonaccorsi[1], Lucie Delemotte[2]\*, David Drew[1]\***

[1]Department of Biochemistry and Biophysics, Science for Life Laboratory, Stockholm University, Stockholm, Sweden; [2]Department of Applied Physics, Science for Life Laboratory, KTH Royal Institute of Technology, Stockholm, Sweden

**\*For correspondence:**
david.drew@dbb.su.se (DD);
lucie.delemotte@scilifelab.se (LD)

**Abstract** In mammals, glucose transporters (GLUT) control organism-wide blood-glucose homeostasis. In human, this is accomplished by 14 different GLUT isoforms, that transport glucose and other monosaccharides with varying substrate preferences and kinetics. Nevertheless, there is little difference between the sugar-coordinating residues in the GLUT proteins and even the malarial *Plasmodium falciparum* transporter *Pf*HT1, which is uniquely able to transport a wide range of different sugars. *Pf*HT1 was captured in an intermediate 'occluded' state, revealing how the extracellular gating helix TM7b has moved to break and occlude the sugar-binding site. Sequence difference and kinetics indicated that the TM7b gating helix dynamics and interactions likely evolved to enable substrate promiscuity in *Pf*HT1, rather than the sugar-binding site itself. It was unclear, however, if the TM7b structural transitions observed in *Pf*HT1 would be similar in the other GLUT proteins. Here, using enhanced sampling molecular dynamics simulations, we show that the fructose transporter GLUT5 spontaneously transitions through an occluded state that closely resembles *Pf*HT1. The coordination of D-fructose lowers the energetic barriers between the outward- and inward-facing states, and the observed binding mode for D-fructose is consistent with biochemical analysis. Rather than a substrate-binding site that achieves strict specificity by having a high affinity for the substrate, we conclude GLUT proteins have allosterically coupled sugar binding with an extracellular gate that forms the high-affinity transition-state instead. This substrate-coupling pathway presumably enables the catalysis of fast sugar flux at physiological relevant blood-glucose concentrations.

## Editor's evaluation

The current manuscript investigates the energy landscape of the mammalian sugar porter GLUT5 using enhanced molecular dynamics simulations and biochemical assays. The approach generates important insights into the mechanism of GLUT5 conformational change, and into mechanistic diversity among the GLUT sugar porters more generally. The overall strategy is convincing, and the findings will be of interest to the transporter and membrane biology communities.

## Introduction

Glucose transporters (GLUT) facilitate the rapid, passive flux of monosaccharides across cell membranes at physiologically relevant concentrations ranging from 0.5 to 50 mM (*Holman, 2020*; *Mueckler and Thorens, 2013*). In human, most GLUT isoforms transport D-glucose, but with different kinetics, regulation, and tissue distribution (*Holman, 2020*; *Mueckler and Thorens, 2013*). For example, GLUT1 is a ubiquitously expressed transporter with saturation by D-glucose around 5 mM to maintain

blood-glucose homeostasis, whereas the liver isoform GLUT2 is saturated at ~50 mM, enabling a high flux of glucose after feeding-induced insulin secretion (*Holman, 2020*; *Thorens, 2015*). Others, like GLUT4, are localized to intracellular vesicles, but will traffic to the plasma membrane of adipose and skeletal muscle cells in response to insulin signaling (*Huang and Czech, 2007*). GLUT5 is the only member thought to be specific to D-fructose, and is required for its intestinal absorption (*Douard and Ferraris, 2008*; *Kayano et al., 1990*). In this process, glucose is actively absorbed by sodium-coupled glucose transporters, while D-fructose is taken up passively by GLUT5 (*Koepsell, 2020*). GLUT5 must therefore efficiently transport D-fructose at high sugar concentrations ($K_M$ = 10 mM), whilst still maintaining sugar specificity (*Koepsell, 2020*). It is poorly understood how GLUT proteins retain strict sugar specificity and how sugars are able to catalyze large conformational changes when they bind to GLUT proteins with weak (mM) affinities (*Drew et al., 2021*). As gate keepers to metabolic re-programming (*Zhang et al., 2019*; *Ancey et al., 2018*), answers to these fundamental questions could have important physiological consequences for the treatment of diseases, such as cancer and diabetes.

GLUT transporters belong to the Major Facilitator Superfamily (MFS), whose topology is defined by two six-transmembrane (TM) bundles connected together by a large, cytosolic loop (*Figure 1A*; *Drew et al., 2021*). Within the MFS, GLUT proteins belong to a separate subfamily referred to as sugar porters, which are distinct from other well-known sugar transporters such as LacY (*Drew et al., 2021*; *Pao et al., 1998*). Sugar porters are subclassified based on a unique sequence motif (*Maiden et al., 1987*; *Nomura et al., 2015*), and crystal structures reveal that this motif corresponds to residues forming an intracellular salt-bridge network, linking the two bundles on the cytoplasmic side (*Figure 1A*, *Figure 1—figure supplement 1A*; *Nomura et al., 2015*; *Deng et al., 2015*). The salt bridges are formed between the ends of TM segments and an intrahelical bundle (ICH) of four to five helices. Crystal structures of GLUT1 (*Deng et al., 2014*), GLUT3 (*Deng et al., 2015*), and GLUT5 (*Nomura et al., 2015*) and related homologs (*Sun et al., 2012*; *Quistgaard et al., 2013*; *Wisedchaisri et al., 2014*; *Qureshi et al., 2020*; *Paulsen et al., 2019*) have shown that the GLUT proteins cycle between five different conformational states: outward open, outward occluded, fully occluded, inward occluded, and inward open (*Figure 1B*). Whilst GLUT proteins are made up from two structurally similar N-terminal (TM1–6) and C-terminal (7–12) bundles, structures have shown that glucose is not coordinated evenly, but almost entirely by residues located in the C-terminal bundle (*Deng et al., 2015*). In particular, residues in the half-helices TM7b and TM10b make up a large fraction of the sugar-binding site (*Drew et al., 2021*). The current working transport model is that the half-helices TM7b and TM10b undergo local conformational changes in response to sugar binding and control substrate accessibility to its binding site from the outside and inside, respectively (*Figure 1B*; *Drew et al., 2021*; *Nomura et al., 2015*). In brief, upon sugar binding from the outside, the inward movement of the extracellular gating helix TM7b is followed by the outward movement of TM10b, and the subsequent breakage of the cytoplasmic inter-bundle salt-bridge network, enabling the two bundles to move around the substrate (*Drew et al., 2021*). In the inward-facing state, TM10b moves fully away from TM4b, sugar exits, and the protein spontaneously resets itself to the outward-open state (*Figure 1B*). Resetting back through an empty-occluded state is rate limiting and ~100-fold slower than via a loaded-occluded intermediate (*Mueckler and Thorens, 2013*; *Lowe and Walmsley, 1986*).

The intermediate, occluded conformation is arguably the most informative for understanding how sugar binding and transport are ultimately coupled (*Drew et al., 2021*). Due to its transient nature, this state is rarely seen. However, it was fortuitously captured in the recent structure of the malarial parasite *Plasmodium falciparum* transporter *Pf*HT1 (*Qureshi et al., 2020*). Nonetheless, *Pf*HT1 is very distantly related to GLUT proteins (*Figure 1—figure supplement 1*), and while GLUT proteins show strict sugar specificity, *Pf*HT1 transports a wide range of different sugars, making it unclear whether this occluded state would constitute a good representative of occluded state in GLUT proteins (*Qureshi et al., 2020*; *Woodrow et al., 1999*). Somewhat unexpectedly, the glucose-coordinating residues in *Pf*HT1 were found to be almost identical to those in *human* GLUT3 (*Qureshi et al., 2020*; *Jiang et al., 2020*; *Deng et al., 2015*). Based on the position of TM7b and biochemical analysis, it was concluded that the extracellular gating helix had evolved to transport many sugars, rather than the sugar-binding site itself (*Qureshi et al., 2020*). Simply put, it was proposed that *Pf*HT1 was less selective in what sugars it transports as its extracellular gate shuts more easily. Whilst the allosteric coupling between TM7b and the sugar-binding pocket might be more pronounced in *Pf*HT1, we hypothesized that the

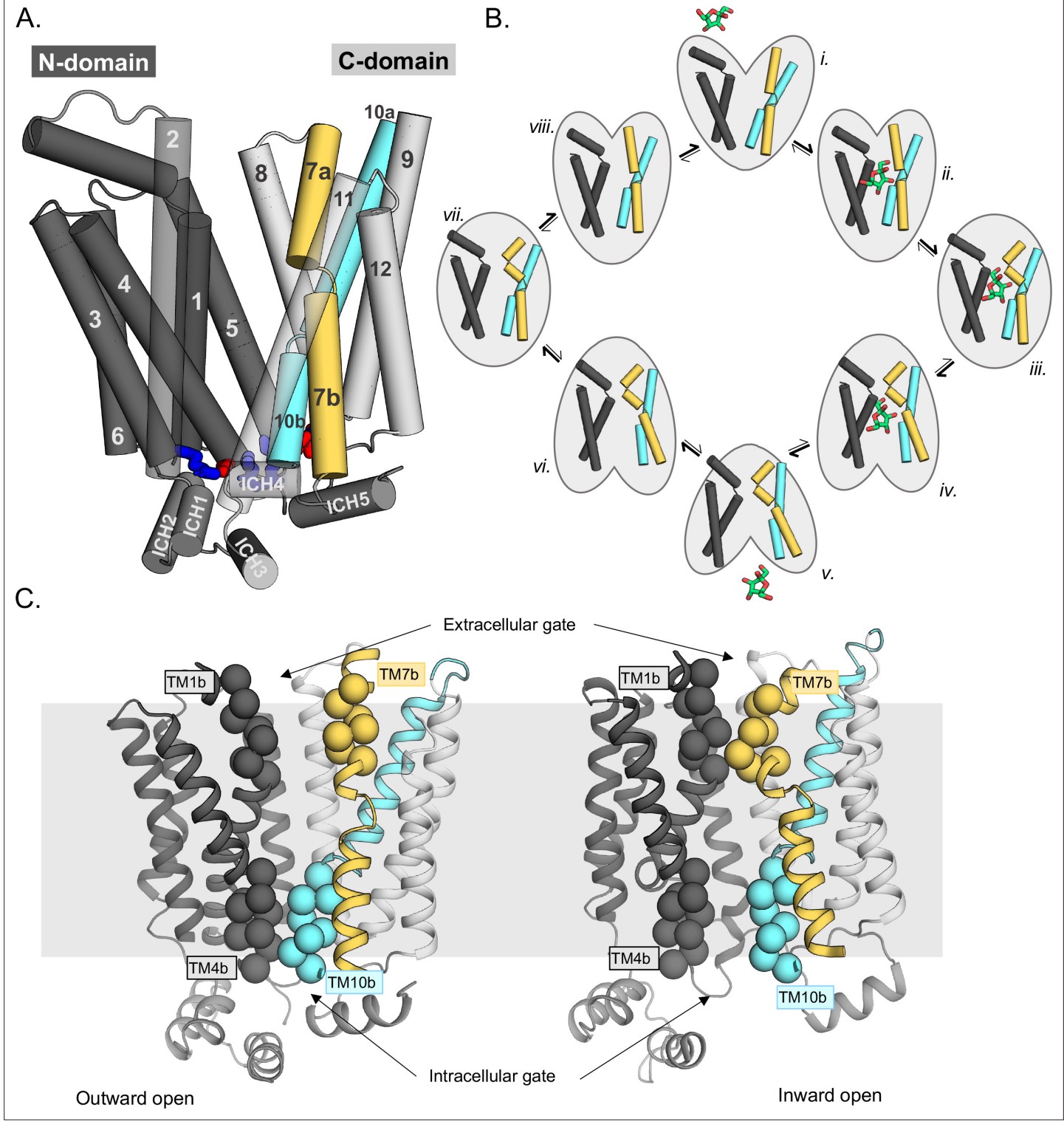

**Figure 1.** Schematic highlighting the GLUT structural features and their major conformations in the transport cycle. (**A**) Structural overview of a sugar porter, GLUT5 (PDB:4ybq). The N-terminal bundle (left, dark gray, transmembrane helices 1–6) and C-terminal bundle (right, light gray, transmembrane helices 7–12) form the separate six-transmembrane bundles, which are connected by the large cytosolic loop comprising intracellular helices (ICH) 1–4. The salt bridge forming residues linking the two bundles are shown as sticks in blue and red, indicating the positive and negative charge of the side chains, respectively. The broken transmembrane helices TM7 (forming TM7a and TM7b) and TM10 (forming TM10a and TM10b) are colored yellow and cyan, respectively. (**B**) Schematic conformational cycle a sugar porter will undergo, based on currently available protein structures. Briefly, moving clockwise from top middle, the transporter will receive a sugar in the outward-open state (PDB:4ybq) (i). The transporter then undergoes a partial

*Figure 1 continued on next page*

*Figure 1 continued*

occlusion of the extracellular gate (outward occluded, ii, PDB:4zw9), followed by full occlusion (iii, PDB:6rw3). Once both gates are fully shut, the inner gates begin to open (inward occluded, iv, PDB:4ja3) where the salt-bridge residues begin to lose contact. Finally, the salt bridges are fully broken apart in an inward-open state (v, PDB:4yb9), and the sugar can be released into the cell. The transporter will then go through the same motions in reversed order in the absence of sugar to reset to the outward-open state (vi–viii). (**C**) The extracellular gate is formed by TM1b and TM7b half-helices, and the intracellular gate is formed by TM4b and TM10b half-helices. Residues defining these gates are shown as spheres. In the outward-open state (left), the extracellular gate is open, and intracellular gate is shut. In the inward-open state (right), the opposite occurs. The gray slab behind the proteins indicates the rough location of the lipid bilayer membrane.

The online version of this article includes the following figure supplement(s) for figure 1:

**Figure supplement 1.** Sequence similarity of rGLUT5 to several GLUTs and model organisms.

fundamental basis for sugar coupling should be conserved in the GLUT proteins (*Drew et al., 2021*; *Qureshi et al., 2020*). Here, using enhanced sampling molecular dynamic simulations and GLUT5-proteoliposome transport assays, we have reconstructed the GLUT5 transport cycle, deciphering the molecular determinants for D-fructose binding and extracellular TM7b gating.

## Results and discussion
### Modeling rat GLUT5 in all conformational states

To piece together the 'rocker-switch' alternating-access mechanism for the GLUT proteins (*Drew and Boudker, 2016*), we must correctly assemble the relevant conformational states along the transport pathway. We thus selected to focus our efforts on the D-fructose transporter GLUT5 for two reasons. Firstly, GLUT5 is the only GLUT protein with structures determined in both outward- and inward-open conformations, which principal component analysis of $n = 17$ sugar porter structures, confirmed represents the two end states (*Qureshi et al., 2020*). Secondly, how non-glucose sugars are recognized by GLUT proteins is unknown, and a computational framework for a D-fructose-specific transporter would help to understand substrate specificity more broadly.

Initially, to fill in the 'missing' GLUT5 conformational states, homology models of *rat* GLUT5 were generated using relevant structures as templates: outward-occluded (*human* GLUT3), occluded

**Table 1.** GLUT5 model and simulation details.

| State modeled | Type of model | ICH5 resolved in template? | Template structure (if applicable) | PDB code | Percentage identity to rat GLUT5 | Time run (ns) | Bilayer used |
|---|---|---|---|---|---|---|---|
| Outward open | Structure | Yes | | 4ybq | 100% | 523 | POPC |
| Outward occluded | Model | Yes | Human GLUT3 | 4zw9 | 40.8% | 298 | POPC |
| Fully occluded | Model | No | *Plasmodium falciparum* hexose transporter (PfHT1) | 6rw3 | 26.0% | 381 | POPC |
| Inward occluded | Model | No | *E. coli* xylose transporter (XylE) | 4ja3 | 23.3% | 158 + (200) | |
| | | | | | | 231 | |
| | | | | | | 188 | POPC |
| Inward open | Model | No | Bovine GLUT5 | 4yb9 | 76.7% | 550 | |
| | | | | | | 144 | |
| | | | | | | 83 | POPC |
| | | | | | | 244 | POPE |
| | | | | | | 232 | 70% POPE, 20% POPG, 10% DOPA |

(*Pf*HT1), and inward-occluded (*E. coli* XylE) and inward-open *bison* GLUT5 (Methods). To assess the stability of the generated homology models, hundreds of nanoseconds-long MD simulations on each of these models were performed in a model POPC membrane bilayer (Methods, *Table 1*). Each model was stable during the simulation, with a slightly higher root mean squared deviation (RMSD) in fully occluded, inward-occluded, and inward-open models, likely reflective of their intrinsic dynamics in absence of substrate as well as ICH mobility when the N- and C-terminal bundles are no longer held together by salt bridges (*Figure 2—figure supplement 1*). Overall, we concluded that the *rat* GLUT5 models had reached an acceptable dynamic equilibrium.

During substrate translocation by MFS transporters, cavity-closing contacts are predominantly formed between TM1 and TM7 on the outside (extracellular gate) and between TM4 and TM10 on the inside (intracellular gate) (*Drew et al., 2021*; *Figure 1C*). We can therefore monitor the distances between the centers of mass (COM) of the residues forming the extracellular gate and intracellular gate as a proxy for the conformational states sampled during simulations (*Figure 2A*). As seen in *Figure 2B*, although the most populated gating distances deviate from the starting GLUT5 models (shown as filled circles), all states equilibrated with mostly overlapping distributions. Notably, the largest deviation from the starting template is for the GLUT5 outward-occluded state modeled from *human* GLUT3, which we attribute to the fact the extracellular gate TM7b was stabilized in the crystal structure by the crystallization lipid monoolein (*Deng et al., 2015*). The non-filled gap between the occluded and inward-occluded state distributions corresponds to the larger global 'rocker-switch' rearrangements (*Figure 2A*), which are inaccessible over these short hundreds of nanoseconds-long time scales.

## Interpolating between models of states using targeted molecular dynamics simulations

Due to the large conformational changes utilized by GLUT proteins, we were unable to connect states by equilibrium-based simulations (Methods). To fully sample the conformational space between the occluded and inward-occluded GLUT5 states, enhanced sampling simulations are necessary. As a first step, we chose here targeted molecular dynamics (TMD) simulations to interpolate between all five major states (*Schlitter et al., 1994*), applying a moving harmonic potential restraint to all heavy atoms in GLUT5. We attempted to apply machine learning methods selection for the selection of collective variables (CVs), as well as several steering methods using said CVs but we were unable to obtain proper state transitions or free energy surface (FES) convergence using these approaches (Methods). After a considerable number of attempts, we decided to select CVs for the string-of-swarms method based on the gating helices TM7 and TM10, which structural studies had shown were important to the transport cycle (*Nomura et al., 2015*; *Deng et al., 2015*; *Qureshi et al., 2020*) and to use TMD to generate an initial pathway interpolating between states (a more detailed discussion of this can be found in Methods and *Table 2*). Nevertheless, given the uncertainty of the *Pf*HT1 occluded state as a suitable model in GLUT proteins, we performed TMD from the outward- to the inward-occluded state either via the occluded state model, or directly between these two states, both with and without D-fructose present. In both these TMD simulation protocols, we find that GLUT5 passes through a conformation in which the positioning of the gates closely matched the ones in the occluded model based on *Pf*HT1 (*Figure 2C*, *Figure 2—figure supplement 2A*).

Having confirmed that the *Pf*HT1 structure is a reasonable approximation for the occluded state in GLUT5, we aimed to characterize the most probable transition pathway linking the outward- and inward-open states, and calculate the FES lining this pathway. To this end, we used the string-of-swarms method for GLUT5 both in its apo (rGLUT5$^{empty}$) and D-fructose-bound (rGLUT5$^{fructose}$) conditions (*Fleetwood et al., 2020*). In brief, each of the five different structural models was represented as beads along a string, with a further 11 beads added from configurations extracted from the TMD, yielding in total 16 beads spanning a tentative initial pathway defined in terms of their intra- and extracellular gate distance (*Figure 2D*, *Figure 2—figure supplement 2B*). From each of these beads, many short trajectories were launched (swarms) to iteratively seek an energy minimum along the string path (see Methods). In this approach, the string simulations converge when the string diffuses around an equilibrium position. This protocol has proven effective to sufficiently explore computational space for complex conformational changes (*Fleetwood et al., 2020*). After ~100 iterations, the strings had converged (*Figure 2—figure supplement 2C*), indicating that an equilibrium position was found.

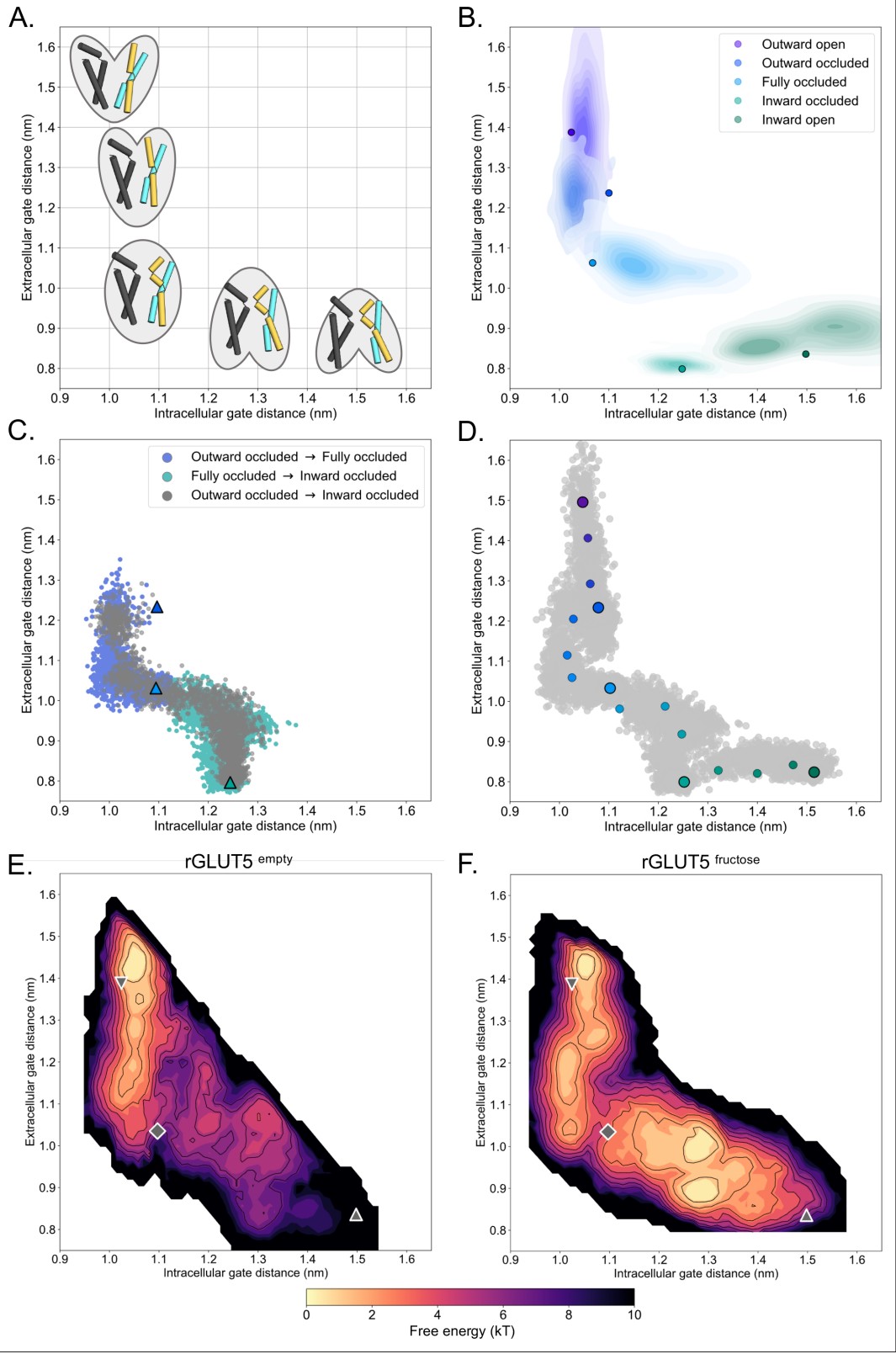

**Figure 2.** A free energy landscape for D-fructose influx by GLUT5. (**A**) A graphical illustration of the five major states of sugar porters, with the intracellular (IC) and extracellular (EC) gate distances on the *x*- and *y*-axis, respectively. Only TM1, TM4, TM7, and TM10 are drawn here, with the intent to highlight only major elements of the rocker switch conformational change. (**B**) IC and EC gate population densities of atomistic simulations of each

*Figure 2 continued on next page*

*Figure 2 continued*

rGLUT5 homology model. Filled circles represent the starting configurations from each rGLUT5 homology model. (**C**) Targeted molecular dynamics (TMD) with bound D-fructose. Individual states are shown in triangles, with the following color schemes: outward occluded: deep blue, occluded: light blue, and inward occluded: green. Gray dots represent all frames corresponding to TMD of outward occluded to inward occluded, skipping the occluded state. This follows a pathway similar to sequential TMD from outward occluded to fully occluded (blue circles), and fully occluded to inward occluded (teal circles). rGLUT5^empty TMD results can be found in ***Figure 2—figure supplement 2A***. (**D**) Beads chosen for the string simulations from TMD projected onto the space defined by the IC and EC gate distances for rGLUT5^fructose. The cloud of gray dots represents all gate distance configurations through the TMD simulations, larger colored dots represent each initial homology model, and the smaller colored dots represent the beads between each of these models, which were chosen for the first iteration of the string-of-swarms method. Beads for rGLUT5^empty found in ***Figure 2—figure supplement 2B***. (**E**) Free energy surface for rGLUT5^empty. The triangles and diamond illustrate the gate measurements for the following homology models: outward open (top left triangle, structure of rGLUT5, PDB:4ybq), fully occluded (diamond, model of PfHT, PDB:6rw3), and inward open (bottom right, model of bGLUT5, PDB:4yb9). (**F**) Free energy surface for rGLUT5^fructose, with the same homology model projection as in **E**.

The online version of this article includes the following figure supplement(s) for figure 2:

**Figure supplement 1.** Root mean squared deviation (RMSD) of atomistic simulations of rGLUT5 homology models.

**Figure supplement 2.** Details of molecular dynamics (TMD), string simulation setup, and convergence.

**Figure supplement 3.** Control simulation distribution along energy surfaces.

**Figure supplement 4.** Projection of free energy surfaces onto alternate collective variable (CV) pairs.

**Figure supplement 5.** Control simulation distribution along energy surfaces.

**Figure supplement 6.** Control simulation distribution along free energy surfaces.

**Figure supplement 7.** Transformation of CV distribution to MSM calculations.

**Figure supplement 8.** Statistical error of free energy surfaces.

**Figure supplement 9.** Average local energy minima conformation superimposed on homology models.

**Figure supplement 10.** Violin plots of alternative collective variables (CVs) calculated for conformations extracted from the D-fructose-bound free energy landscape local minima.

**Figure supplement 11.** Calculation of features along energy surfaces.

---

Nevertheless, we continued to run another ~450–650 iterations to ensure exhaustive sampling of the entire transition pathway, enabling an appropriate estimation of the free energy along the converged path.

## The free energy landscape of GLUT5 with and without d-fructose bound

Once the strings had converged and equilibrium was reached, we calculated FESs based on the transitions of all equilibrium swarm simulations (see Methods). Comparing the FESs between these two conditions reveals obvious differences between rGLUT5^empty and rGLUT5^fructose simulations (***Figure 2E, F***). In the absence of D-fructose, the outward-open state is the most energetically favorable, with higher energy barriers to either occluded or inward-facing states (***Figure 2E***). This calculation is consistent with experimental observations for the related XylE transporter, for which the outward-facing state is the most populated in the absence of sugar (***Jia et al., 2020***). The FES of GLUT5^empty is also consistent with structures that have shown that the strictly conserved salt-bridge network is only present on the cytoplasmic inside (***Drew et al., 2021***), stabilizing the outward-facing state; single-point mutations to the salt-bridge network residues have indeed been shown to arrest GLUT transporters in the inward-facing conformation (***Deng et al., 2014***; ***Schürmann et al., 1997***). In the presence of sugar, the inward-facing states become accessible and are of similar energetic stability to the outward-facing states (***Figure 2F***). The heights of the free energy barriers between outward and inward-facing states in presence and absence of substrate, respectively, are consistent with measurements of GLUT kinetics, as rates have been shown to be 100-fold faster for substrate bound than for empty-occluded transitions (***Mueckler and Thorens, 2013***; ***Lowe and Walmsley, 1986***). In other words, we can directly see the effect of substrate binding on sugar-induced conformational changes.

**Table 2.** Summary of targeted MD simulations.

**rGLUT5$^{empty}$ Outward open – Inward open (condition 1)**

| TMD number | Starting configuration | Target state | Time (ps) | Final heavy atom RMSD (nm) |
|---|---|---|---|---|
| 1.1 | Out open structure | Outward occluded | 11,220 | 0.048415 |
| 1.2 | TMD 1.1 final frame | Fully occluded | 10,890 | 0.050022 |
| 1.3 | TMD 1.2 final frame | Inward occluded | 8120 | 0.062960 |
| 1.4 | TMD 1.3 final frame | Inward open | 10,780 | 0.052656 |
| 1.5 (skipping occluded state validation) | TMD 1.1 final frame | Inward occluded | 10,440 | 0.051464 |

**rGLUT5$^{empty}$ Inward open – Outward open (condition 2)**

| TMD number | Starting configuration | Target state | Time (ps) | Final heavy atom RMSD (nm) |
|---|---|---|---|---|
| 2.1 | In open homology model | Inward occluded | 9620 | 0.066614 |
| 2.2 | TMD 2.1 final frame | Fully occluded | 8620 | 0.053146 |
| 2.3 | TMD 2.2 final frame | Outward occluded | 7760 | 0.058147 |
| 2.4 | TMD 2.3 final frame | Outward open | 9120 | 0.054561 |
| 2.5 (skipping occluded state validation) | TMD 2.1 final frame | Outward occluded | 11,840 | 0.048901 |

**rGLUT5$^{fructose}$ Outward open – Inward open (condition 3)**

| TMD number | Starting configuration | Target state | Time (ps) | Final heavy atom RMSD (nm) |
|---|---|---|---|---|
| 3.1 | Out open structure with D-fructose bound | Outward occluded | 8500 | 0.065881 |
| 3.2 | TMD 3.1 final frame | Fully occluded | 10,780 | 0.052316 |
| 3.3 | TMD 3.2 final frame | Inward occluded | 12,630 | 0.059851 |
| 3.4 | TMD 3.3 final frame | Inward open | 10,260 | 0.045643 |
| 3.5 (skipping occluded state validation) | TMD 3.1 final frame | Inward occluded | 12,480 | 0.055612 |

**rGLUT5$^{fructose}$ Inward open – Outward open (condition 4)**

| TMD number | Starting configuration | Target state | Time (ps) | Final heavy atom RMSD (nm) |
|---|---|---|---|---|
| 4.1 | In open homology model with D-fructose bound | Inward occluded | 10,000 | 0.075750 |
| 4.2 | TMD 4.1 final frame | Fully occluded | 12,980 | 0.048203 |
| 4.3 | TMD 4.2 final frame | Outward occluded | 11,780 | 0.049837 |
| 4.4 | TMD 4.3 final frame | Outward open | 12,420 | 0.049001 |
| 4.5 (skipping occluded state validation) | TMD 4.1 final frame | Outward occluded | 12,480 | 0.049699 |

In the presence of D-fructose, the occluded state model of GLUT5 is exactly positioned between the two energetically favorable outward- and inward-facing states (*Figure 2F*). Consistent with a transition state, the occluded state model is located on the highest energetic barrier along the lowest energy pathway between the two opposite-facing conformations. Moreover, transition into the occluded state from the outward states is energetically unfavorable for GLUT5 without sugar, but the presence of D-fructose clearly lowers the activation barrier (*Figure 2E, F*). These calculations are also consistent with the fact that GLUT transporters are required to spontaneously reset to the opposite-facing conformation through an empty-occluded transition after sugar release (*Drew et al., 2021*),

that is gate rearrangements, forming intermediate states, must also be able to spontaneously close in the absence of sugar.

To further assess the stability of the conformational states described in the free energy landscape, we further launched unbiased molecular dynamics simulations from various energy wells depicted in *Figure 2F* (see Methods). These simulations tended to drift from their local energy wells (*Figure 2— figure supplement 3*), often not converging into free energy minima. We reasoned that if this drift in the landscape was due to the selection of the CVs, that projections of these controls in other CV spaces should remain in an energy minima. We therefore calculated energy landscapes using other state-defining features as CVs, and subsequently projected the unbiased simulations onto these landscapes (*Figure 2—figure supplements 4 and 5*). The simulations converged into their own or a neighboring energy well in these projections, indicating that the conformations from enhanced sampling are reliable. Taken together, and based on the overall differences in the free energy landscapes between rGLUT5$^{empty}$ and rGLUT5$^{fructose}$ simulations matching experimental observations, we conclude that the free energy landscapes represent a physiologically relevant GLUT5 conformational cycle.

## Conformational stabilization of D-fructose coordination in the occluded state

Since the presence of D-fructose lowers the energetic barriers between outward- and inward-facing states (*Figure 2F*), we reasoned that we should be able to extract the molecular determinants for D-fructose coordination in the occluded state from these simulations. Based on extensive biochemical data and the glucose-bound *human* GLUT3 structure, we know that D-glucose is transported with the C1-OH group facing the bottom of the cavity (endofacial) and the C6-OH group facing the top (exofacial) (*Drew et al., 2021*; *Holman, 2018*; *Barnett et al., 1973*; *Barnett et al., 1975*). It is expected that D-fructose will be likewise transported by GLUT5 with the C1-OH group facing the endofacial direction, since substituents to D-fructose were better tolerated when added to the C6-OH position (*Tatibouët et al., 2000*; *Yang et al., 2002*). D-fructose was unconstrained during TMD and string simulations. To evaluate the conformational heterogeneity of D-fructose, we binned the energy landscape, extracted configurations corresponding to each bin, and then clustered the D-fructose poses for these ensembles of configurations (see Methods). As seen in *Figure 3A*, in the outward-open and outward-occluded conformations, D-fructose does not display any preferential binding mode, and the C1-OH group has no preferential orientation (brown sphere). In contrast, in the occluded state, the sugar becomes highly coordinated, adopting a single well-defined binding pose in approximately 65% of conformations that is ninefold more populated than the next most abundant pose (*Figure 3A, B*, Methods, *Table 3*). Remarkably, the two most preferred poses are very similar to the orientation that both D-glucose and D-xylose have adopted in previously determined crystal structures (*Figure 3C*; *Drew et al., 2021*; *Deng et al., 2015*; *Sun et al., 2012*).

Upon closer inspection of the D-fructose-bound state, we see that the serine residue S391, located between the TM10a-b breakpoint, coordinates the C1-OH group of D-fructose (*Figure 3B, D*), which is a glycine residue in the glucose-specific GLUT members (*Figure 1—figure supplement 1*). The mutation of S391 to alanine in *rat* GLUT5 has been shown to weaken D-fructose binding (*Nomura et al., 2015*). As seen in D-glucose-bound sugar porter structures (*Deng et al., 2015*; *Sun et al., 2012*; *Qureshi et al., 2020*; *Jiang et al., 2020*; *Bavnhøj et al., 2021*), the strictly conserved TM5 glutamine Q166 is also forming tight hydrogen-bond interactions to the C1-OH group of D-fructose (*Figure 3B, D*). Hydrogen bonding to Q166 is thought to be a critical interaction required for transport, as it is the only N-terminal bundle residue seen to coordinate D-glucose in sugar-bound structures (*Drew et al., 2021*; *Deng et al., 2015*). Unexpectedly, the TM7a glutamine residues Q287 and Q288 coordinating the C2- and C3-OH groups of D-glucose in GLUT3 and other members (*Figure 1—figure supplement 1*; *Drew et al., 2021*), are not forming direct hydrogen bonds with D-fructose in the GLUT5 simulations (*Figure 3B, D*). We find, however, that the most highly coordinated waters in the MD simulations are located between Q287 and Q288 and D-fructose, which are within hydrogen-bond distance to the C2-OH and C3-OH groups (*Figure 3B*). In GLUT1, GLUT3, and GLUT4 structures, the bulky TM10 tryptophan residue directly coordinates the C1-OH group of D-glucose (*Deng et al., 2015*; *Deng et al., 2014*; *Yuan et al., 2022*). GLUT5, however, lacks the TM10 tryptophan, and this residue is replaced by A395 (*Figure 3B*). It appears that S391 in TM10 is instead forming the C1-OH hydrogen-bond interaction, which was unexpected. The interaction between the S391 in TM10 and the C1-OH

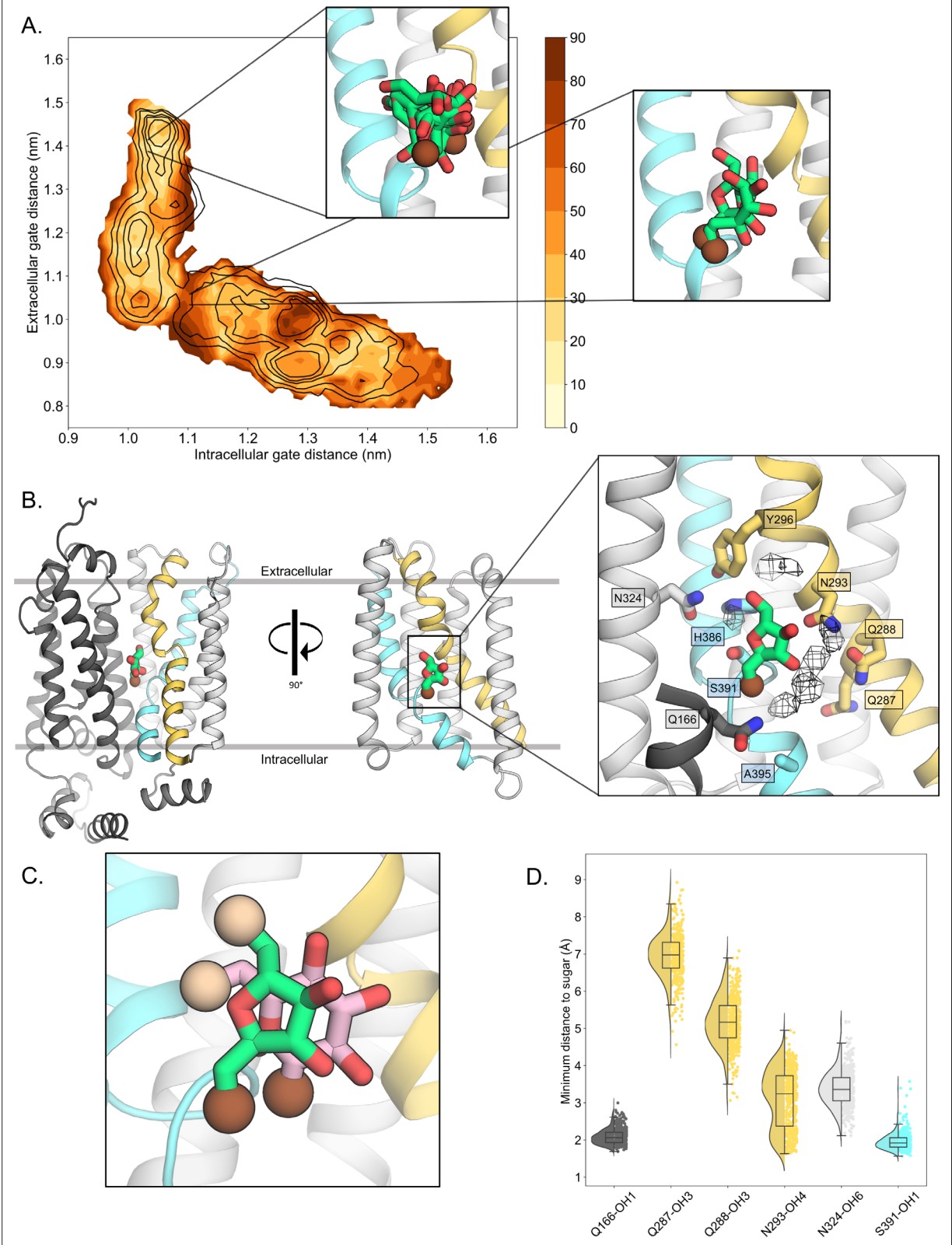

**Figure 3.** D-fructose becomes highly coordinated in the occluded conformation. (**A**) Sugar coordination of the D-fructose-bound simulations superimposed onto the free energy landscape for rGLUT5$^{fructose}$, colored according to the frequency of the most populated cluster in each bin, therefore darker colors indicate a more consistent pose (see Methods for bin description). Snapshots extracted from two bins, corresponding to the outward-open or occluded states, respectively, depicting ~70% of total pose variability (see **Table 3** in Methods) are shown as inserts. GLUT5 helices, D-fructose,

*Figure 3 continued on next page*

*Figure 3 continued*

and C1 hydroxyl group shown as in panel B. (**B**) An overview of a representative binding pose of D-fructose in occluded GLUT5, colored as in *Figure 1*. The D-fructose C1 hydroxyl group is colored as a brown sphere for orientation purposes. Selected interacting residues shown as sticks, and the density of waters near the D-fructose in the occluded state is shown as a black mesh. (**C**) The coordination of D-fructose is oriented the same as previously determined glucose positions, such as in PfHT1 (PDB:6rw3, pink). The C6 hydroxyl group is shown as an enlarged sphere colored tan, and the C1 hydroxyl group in brown as in panel B. The D-fructose pose chosen here is the most populated cluster, as seen in Methods, *Table 3*. (**D**) The distribution of distances of indicated D-fructose hydroxyl groups to certain side chains. Distances shown are calculated from the most populated cluster in the occluded state bin (see *Table 3*).

The online version of this article includes the following figure supplement(s) for figure 3:

**Figure supplement 1.** The coordination of N293 (licorice) through the transport cycle.

moiety means that Q287 side chain would be too far away to directly coordinate the C2-OH group of D-fructose. Although this would make the C2-OH less well coordinated, this reasoning is consistent with biochemical analysis. Specifically, GLUT5 can transport 2,5-dihydromannitol with similar kinetics to D-fructose, which is a sugar that is identical to the furanose form of D-fructose, but lacks the C2-OH group (*Yang et al., 2002*). In contrast, D-fructose epimers that differ by the orientation in any of the other –OH groups are thought to be unable to bind, since they do not display any cold competition for [14]C-D-fructose uptake when added to cells expressing *human* GLUT5 (*Yang et al., 2002*).

Upon superimposition of all the major conformations along the GLUT transport cycle, the TM7b asparagine (N293) was shown to be the only sugar-coordinating residue significantly changing its position during the transport cycle (*Qureshi et al., 2020*; *Figure 3—figure supplement 1*). Because the TM7b asparagine residue is strictly conserved in all GLUT transporters and related sugar porters, it was proposed that the recruitment of the TM7b asparagine is a key and generic interaction required for coupling sugar binding with extracellular TM7b gating (*Drew et al., 2021*). Consistently, in the simulations, the TM7b asparagine (N293) is also well positioned to coordinate the C4-OH group of D-fructose in the fully occluded state, and generally maintains hydrogen-bond distance (*Figure 3B, D*). In addition to the TM7b asparagine (N293), a TM7b gating tyrosine (Y296) also forms an interaction to a histidine residue (H386) in TM10a (*Figure 4A*). Both the TM10a histidine (H386) and the TM7b tyrosine (Y296) are unique to GLUT5 (*Figure 1—figure supplement 1*; *Nomura et al., 2015*) and GLUT5 variants H386F, H386A, and Y296A have been shown to severely diminish D-fructose binding (*Nomura et al., 2015*). The TM7b tyrosine (Y296) appears to also interact with an asparagine residue (N324) (*Figure 4A*), which is also generally within hydrogen-bond distance to the C6-OH group of D-fructose (*Figure 3D*). As such, the TM7b gate seems to be connected both indirectly and directly to the sugar-binding site.

**Table 3.** Analysis of sugar-binding pose in outward-open and occluded states.

| Bin closest to: outward-open state<br>Total frames in bin: 1000<br>Total possible clusters for sugar pose: 217 | | | Bin closest to: occluded state<br>Total frames in bin: 690<br>Total possible clusters for sugar pose: 116 | | |
|---|---|---|---|---|---|
| Cluster number | Number of frames in cluster | Percentage of total frames in bin | Cluster number | Number of frames in cluster | Percentage of total frames in bin |
| 1 | 234 | 23.40% | 1 | 447 | 64.78% |
| 2 | 123 | 12.30% | 2 | 52 | 7.54% |
| 3 | 106 | 10.60% | 3 | 24 | 3.48% |
| 4 | 86 | 8.60% | 4 | 23 | 3.33% |
| 5 | 59 | 5.90% | 5,6,7 | 3 | 0.43% |
| 6 | 38 | 3.80% | 8–12 | 2 | 0.29% |
| 7 | 36 | 3.60% | 13–116 | 1 | 0.14% |
| 8–12 | 32–10 | 3.20–1.00% | | | |
| 13–16 | 5–2 | 0.50–0.20% | | | |
| 17–217 | 1 | 0.10% | | | |

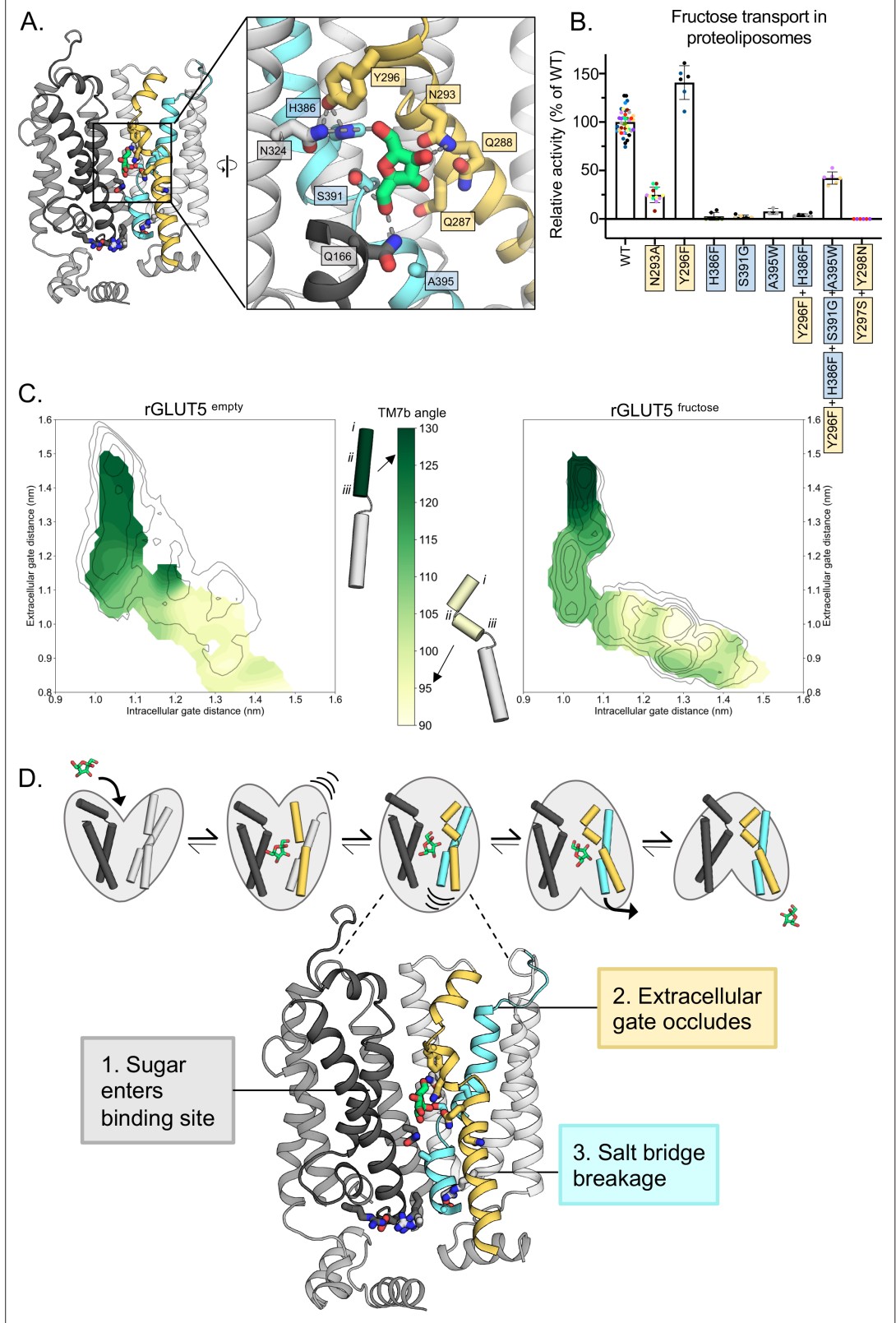

**Figure 4.** Confirming the residues required for coupling sugar binding to the extracellular gate TM7b and the intracellular gate TM10b. (**A**) An overview of the residues connecting the coupling of TM7b breakage during sugar binding to the movement of TM10b. The occluded state is represented here. When N293 is pulled toward the D-fructose hydroxyl group on C4, TM7b is kinked, and Y296 is rotated toward TM10. Y296 is then able to interact

*Figure 4 continued on next page*

*Figure 4 continued*

with H386 on TM10a. Coloring and objects shown as described in *Figure 1*. TM2, TM11, and ICH4 are omitted for clarity, though the salt-bridge residue R407 on TM11 remains. The inlay highlights key residue interactions with the sugar, the Y296–H386 interactions, and the Y296–N324 interactions. Dashes represent possible hydrogen-bond interactions (as shown in *Figure 3B, D*, not including Y296–H386 or Y296–N324). (**B**) $^{14}$C-D-fructose transport activity of wild-type (WT) rGLUT5 and mutations related to the coupling of D-fructose binding with TM7 breakage and TM10 movement, residue names are colored as in panel A. Uptake of $^{14}$C-D-fructose (black bars) in proteoliposomes of rat GLUT5 variants relative to WT. Errors bars represent standard error of the mean (SEM) of n ≥ six independent measurements taken from two to three independent liposome reconstitutions. Each independent reconstitution is colored differently. Colors shared between different bars indicate shared liposome preparation prior to reconstitution. (**C**) TM7b angle for rGLUT5$^{empty}$ (left) and rGLUT5$^{fructose}$ (right) superimposed onto the respective free energy landscapes. This angle is calculated by measuring the vector formed between residue groups *i*, *ii*, and *iii* as shown on the protein cartoons (see Methods). The angle in the outward-open state is approximately 130 degrees, and will bend to nearly a 90 degree angle toward the inward-facing states. (**D**) Our updated view on the molecular determinants of sugar-induced conformational change in GLUT5. Residues moving to interact with the sugar in the binding site induce a conformational change in TM7b (yellow), occluding the extracellular gate. This breakage of TM7b in a fully occluded state is communicated to an inward opening via TM10b (cyan). Interactions between TM7b (Y296) to TM10a (H386) and TM8 (N324), as well as between the D-fructose and S391, stabilize TM10a. The TM10b, which is separated from TM10a by the GPXPXP motif, is uncoupled from TM10a, and will move away from TM4b. This movement will then induce breakage of the salt-bridge residues to an inward-open state. Coloring as in *Figure 4—source data 1* for 4B is supplied.

The online version of this article includes the following source data and figure supplement(s) for figure 4:

**Source data 1.** Raw data from sugar transport assays of GLUT5.

**Figure supplement 1.** GLUT5 kinetics and size exclusion chromatography (SEC) traces for purified rGLUT5 proteins.

**Figure supplement 2.** Sequence logos for transmembrane helices of putative GLUT5 proteins (see Methods) in other organisms.

**Figure supplement 3.** Salt-bridge formation during the conformational cycle.

---

To strengthen the proposed MD model for D-fructose coordination, we evaluated $^{14}$C-D-fructose transport of purified GLUT5 variants reconstituted into liposomes using our recently developed transport assay (*Saudea et al., 2022*) (see Methods for details). We focused the mutagenesis on GLUT residues that are different in GLUT5 from GLUT transporters unable to transport D-fructose (*Li et al., 2004*). Consistently, the H386F and S391G variants that were constructed to match the most common residue in the other GLUT transporters (*Deng et al., 2015*), abolished D-fructose transport (*Figure 4B*). As shown recently (*Saudea et al., 2022*), the A395W variant is also incapable of D-fructose transport (*Figure 4B*). All GLUT5 variants were unable to transport D-glucose (*Figure 4—figure supplement 1C*). The H386F and S391G mutations are particularly informative, as the corresponding residues do not participate in coordination of D-glucose in the glucose-bound GLUT structures (*Deng et al., 2015*; *Deng et al., 2014*; *Yuan et al., 2022*). Confirming the importance of the strictly conserved TM7b asparagine N293, its substitution to alanine also severely reduced transport (*Figure 4B*).

## Coupling between D-fructose binding and TM7b gating rearrangements

In the *Pf*HT1 occluded structure, TM7b was found to have broken into an elbow-shaped helix, with close contacts to TM1 at the breakpoint. In all inward-facing states, the gating helix TM7b remains at a sharp angle (*Drew et al., 2021*). Structural information suggests that the stabilization of TM7b via the asparagine coordination to a substrate sugar would induce TM7b to transition from a bent to a broken-helix conformation (*Drew et al., 2021*; *Qureshi et al., 2020*). To assess this structural transition, we compared the angle formed by TM7b throughout both rGLUT5$^{empty}$ and rGLUT5$^{fructose}$ simulations (*Figure 4C*). Consistently, we observe that TM7b comparatively forms a sharper angle earlier in the transport cycle when sugar is present, indicating that indeed the conformational state of TM7b is connected with sugar recognition, and suggesting a mechanism whereby D-fructose binding induces transition into the occluded states. The angle of TM7b further decreases upon transition into the occluded state to fully shut the outside gate.

GLUT transporters harbor two strictly conserved tyrosine residues in TM7b (*Nomura et al., 2015*) (Y297 and Y298), which form the substrate occlusion in the outward-occluded conformation (*Drew et al., 2021*). In *Pf*HT1, TM7b residues are considerably more polar and the TM7b tyrosine residues have been replaced by serine and asparagine (*Qureshi et al., 2020*). *Pf*HT1 mutations of TM7b gating residues were found to be just as critical to transport as residues coordinating D-glucose (*Qureshi et al., 2020*). Furthermore, TM7b variants retaining transport competence shifted the substrate preference of *Pf*HT1 from D-glucose toward D-fructose (*Drew et al., 2021*; *Qureshi et al., 2020*). It was, thus, proposed that *Pf*HT1 had not evolved the sugar-binding site to be able to transport many sugars, but its extracellular gate. In MD simulations of GLUT5, the TM7b tyrosine residue Y296 that formed an interaction with H386, is located next to these occlusion-forming Y297 and Y298 residues. In most other GLUT isoforms, Y296 is a phenylalanine, and so to assess this interaction further we measured transport of an Y296F and the double H386F, Y396F variant. Consistent with the H386 required for D-fructose, coordination the double H386F, Y396F variant showed no transport activity (*Figure 4B*, *Figure 4—figure supplement 1A*). Somewhat surprisingly, the single Y296F variant retained robust $^{14}$C-D-fructose transport with an ~40% improved performance ($k_{cat}/K_M$) (*Figure 4B*, *Figure 4—figure supplement 1A*). Upon closer inspection, we find that the TM7b tyrosine is a phenylalanine residue in around 30% of more distantly related GLUT5 homologs (*Figure 4—figure supplement 2*). Indeed, independent co-evolution analysis corroborates that Y296 and H386 residues are forming an interaction in the occluded state of GLUT5 (*Mitrovic et al., 2022*). Moreover, in yeast-based forward-evolution screens of hexose transporters, the equivalent residue to Y296 in TM7b was the only single-residue-variant uncovered with a shifted substrate preference from D-glucose toward D-xylose (*Wang et al., 2016b*; *Wang et al., 2016a*). It is possible that the divergence of the TM7b residue from phenylalanine to tyrosine has evolved so that GLUT5 can transport an as yet, unidentified sugar in some tissues, which has been speculated previously, since GLUT5 is expressed in tissues with very low circulating levels of D-fructose (*Douard and Ferraris, 2008*).

To further demonstrate the importance of coupling between TM7b and the sugar-binding site, we combined single-point mutations that individually abolished D-fructose transport (H386F, S391G, and A395W) with the TM7b variant (Y296F). Interestingly, the combined variant is now re-able to transport D-fructose transport with ~40% wildtype activity (*Figure 4B*, *Figure 4—figure supplement 1B*), although it is still unable to transport D-glucose (*Figure 4—figure supplement 1C*). Given the already low affinity (10 mM) of D-fructose for wildtype GLUT5, we are unable to measure kinetics of this combined variant under the current experimental setup (*Saudea et al., 2022*). Nonetheless, the much higher accumulation observed above background indicates multiple turnover events. Thus, whilst the GLUT5-binding site has evolved to specifically coordinate D-fructose, there is enough plasticity in the sugar-binding pocket that can allow for alternative interactions when combined with differences to extracellular TM7b gating residues, that is, as was recently concluded by analysis of the malarial parasite transporter structure *Pf*HT1 (*Qureshi et al., 2020*). So far, however, fully shifting sugar preferences from the natively preferred substrate to a different sugar remains elusive, and it is unlikely that mutations to the TM7b gating region are enough. For example, GLUT5 variants of the gating tyrosine Y297N and Y298S, constructed to match the more polar TM7b in the promiscuous transporter in *Pf*HT1, abolished D-fructose transport (*Figure 4B*). We expect that variants to influence global bundle dynamics, in addition to extracellular gating differences, are likely required.

Extracellular TM7b gate closure in the occluded conformation must somehow trigger the breakage of the intracellular salt-bridge network on the inside in order for the two bundles to come apart. The mutational analysis of sugar-binding residues supports the modeled D-fructose, which implies that the substrate sugar stabilizes both the closure of TM7b as well as an interaction with TM10a. MD simulations shows that when TM10a becomes locked in place by interaction with the substrate sugar, TM10b is able to move more independently (*Figure 4—figure supplement 3A, B*), which is facilitated by a very mobile GPXPXP helix-break motif (*Figure 1—figure supplement 1*, *Figure 4—figure supplement 2*). In the simulations, we see that the TM7b angle decreases from about 140 degrees to about 115 degrees without any noticeable change in TM10b (*Figure 4—figure supplement 3A*). However, as the TM7b angle reaches 115 degrees in the occluded state, TM10b undergoes a large shift in position. Interestingly, the salt-bridge residues, particularly those located between TM4 and TM11, do not fully break apart until TM10b has finished rearranging (*Figure 4—figure supplement 3B, C*). This

would indeed be consistent with the coordinated coupling between the inward movement of TM7b triggering the outward movement of TM10b to break the inter-bundle salt-bridge network.

## Discussion

GLUT transporters are often presented as text-book examples of how small molecule transporters are functional equivalents of soluble enzymes. Yet, despite extensive kinetic, biochemical, and physiological analysis, we have a poor understanding of how GLUT structures fit into such a molecular description. Here, for the first time, we can confirm that the occluded state structure of *Pf*HT1 (*Qureshi et al., 2020*) provides a suitable template for modeling the transition state in a GLUT transporter. The classical description of enzyme catalysis is that there is relatively weak binding of the substrate to the enzyme in the initial state, but a tight binding in the transition state (*Henzler-Wildman and Kern, 2007*; *Klingenberg, 2005*). This conceptual framework implies that in GLUT proteins the sugar would bind more tightly to the transition state, which would be consistent with the Induced Transition Fit of transport catalysis proposed by *Klingenberg, 2005*. More specifically, in the occluded state, we find that TM7b is broken over the sugar-binding site to better coordinate D-fructose. The fundamental difference between enzymes and transporters is that the structure of the transition state determines the activation barrier for global conformational changes in transporters, whereas in enzymes, the barrier is imposed by substrate remodeling in the transition state (*Klingenberg, 2005*). Here, we indeed observe that the energy barrier for conversion between states is clearly lowered by the coordination of D-fructose. It is worth mentioning, that we have focused on the role of the sugar coupling in influx rather than efflux, because the affinities for D-glucose are reported to be 10-fold lower on the inside in other GLUT transporters and homologs (*Drew et al., 2021*; *Cloherty et al., 1996*) and salt-bridge formation between the two bundles was more difficult to model (see Methods, *Figure 4—figure supplement 3D*).

By measuring GLUT1 kinetics at different temperatures, an activation barrier (Ea) of around 10 kcal/mol has been reported (*Lowe and Walmsley, 1986*). This relatively low activation barrier roughly corresponds to the breakage of a few salt bridges, which matches the expectation for the intracellular salt-bridge-rich GLUTs. The D-glucose-binding energies have been estimated to be around 9 kcal/mol for GLUT3 (*Liang et al., 2018*), which is consistent with sugar binding required to generate the global transitions by inducing formation of the occluded state. Although the transition state represents the highest energetic barriers between opposite-facing conformations in MD simulation of GLUT5, the height of the activation barrier cannot be reliably calculated from our simulations for several different reasons. Firstly, the energy barriers are estimated along a path that describes structural transitions in the extra- and intracellular gates, rather than all conformational changes across the entire protein, that is, whilst the CVs allowed us to explore the entire transport pathway, different CVs that more accurately separate metastable states are required, which we have explored in an accompanying work using co-evolution analysis (*Mitrovic et al., 2022*). Secondly, our models consider a membrane bilayer made from POPC lipids, whereas it is well established that transport by GLUT proteins requires the presence of anionic lipids (*Saudea et al., 2022*; *Hresko et al., 2016*). The fact that the activation barrier for GLUT1 has been shown to increase from 10 to 16 kcal/mol in liposomes made from lipids with longer fatty acids highlights just how sensitive GLUT proteins are to the lipid composition (*Carruthers and Melchior, 1984*). Here, we chose to use a neutral lipid composition to avoid complications related to the anticipated time scales needed to equilibrate a complex bilayer. Importantly, the coordination of D-fructose in the occluded state is consistent with previous biochemical analysis, the coordination of D-glucose, and the GLUT5-proteoliposome assays carried out here.

The asparagine residue located in the beginning of TM7b (N293) is conserved across the entire sugar porter superfamily, and is the only sugar-coordinating residue significantly changing its position during the transport cycle (*Drew et al., 2021*; *Qureshi et al., 2020*). Our simulations show that N293 is recruited to coordinate D-fructose in a similar manner to D-glucose, and we propose that the TM7b asparagine N293 is a generic interaction required for coupling the binding of any transported monosaccharide sugar with the extracellular TM7b gate. Indeed, all TM residues in *human* GLUT1 has been substituted to cysteine in a cysteine-less GLUT1 variant, which has wildtype activity in oocytes (*Mueckler and Makepeace, 2009*). The mutation of the TM7b asparagine in *human* GLUT1 to cysteine was one of just a few residues showing less than 10% wild-type (WT) activity. It is worth noting that in the

proteoliposome setup the N293A mutant in GLUT5 is not completely inactive (*Figure 4B*), and we are investigating whether the TM7b gate for this mutant can still (inefficiently) close under efflux conditions, that is the signal would therefore be accumulation from oppositely orientated GLUT5.

Once TM7b has been stabilized by a substrate sugar via N293, the closure of TM7b is influenced by residues along the length of TM7b. In the promiscuous sugar transporter *Pf*HT1, the strictly conserved TM7b gating tyrosine residues found in GLUT1–14, have been replaced by serine and asparagine (*Figure 1—figure supplement 1*; *Qureshi et al., 2020*). These more polar residues enable closing of the outside gate more easily and play a role in catalyzing transport of different sugars. In contrast, GLUT5 is a highly specific sugar transporter and has a finely tuned extracellular TM7b gate. More specifically, upon TM7b gate closure, the tyrosine residue Y296 preceding the 'YY' motif forms a unique pairing to a histidine residue peripheral to the sugar-binding site (H386). The histidine can interact both with the TM7b tyrosine as well as hydrogen bond to an asparagine residue interacting with D-fructose (N324). It is critical that we observe a connection between the sugar-binding site and the TM7b tyrosine residue, which is located at the region wherein the TM7b helix transitions from a bent to a broken helix in the occluded state. The importance of TM7b was also observed in the *E. coli* xylose symporter XylE. Whilst XylE binds D-glucose in the same manner and with the same affinity as in *human* GLUT3, the XylE protein is incapable of transporting the sugar, that is, D-glucose is a dead-end inhibitor (*Farwick et al., 2014*). However, the mutation of the residue corresponding to the TM7b tyrosine in XylE (L297F) together with a sugar-binding site mutant (Q175L), enables XylE to transport D-glucose while retaining 75% of WT D-xylose transport (*Madej et al., 2014*). Illustrating the importance of the TM7b gate, we have been able to reconstruct a GLUT5 sugar-binding site with individually 'dead' mutants that can recover D-fructose transport when combined with the mutagenesis of an TM7b gating residue. Taken together, our work strengthens the proposal that TM7b should be considered as an extension of the sugar-binding site (*Qureshi et al., 2020*).

## Conclusions

We conclude that the molecular determinants for sugar transport are an intricate coupling between an extracellular gate, a sugar-binding site, and an intracellular salt-bridge network (*Figure 4D*). Weakly binding sugars are able to induce large conformational changes in GLUT proteins by conformational stabilization of a transition state that can already be spontaneously populated. Rather than a substrate-binding site that achieves strict specificity by having a high affinity for the substrate, GLUT proteins have allosterically coupled sugar binding with an extracellular gate that forms the high-affinity transition-state instead. Presumably, this substrate-coupling pathway ensures that sugar binding does not become rate limiting, and so enables GLUT proteins to catalyse fast sugar flux at physiological relevant blood-glucose concentrations in the mM range. The recent type 2 diabetes drug empagliflozin in complex with the sodium-coupled glucose transporter SGLT2, demonstrates how selective inhibition was achieved by the aglycone of the glucoside inhibitor interacting with the mobile TM1a-b and TM6a-b half-helices (*Niu et al., 2022*). In many aspects, while GLUT proteins are referred to as rocker-switch proteins, their asymmetric binding mode gives rise to gating elements closely resembling the structural transitions seen in rocking-bundle proteins, like SGLT2 (*Drew et al., 2021*). Such an intricate coupling indicates that pharmacological control of GLUT proteins might best be accomplished by small molecules targeting gating regions extending from the sugar-binding site.

## Materials and methods
### Protein modeling
Residue numbing for rGLUT5 is based on the UNIPROT entry of rGLUT5: P43427, all generated models begin at residue E7 and end at residue V480. The starting models for rGLUT5 in each state were generated using homology modeling with MODELLER version 10.1 (*Sali and Blundell, 1993*). A summary of the details of these models and subsequent simulations are found in *Table 1*. Details of each state's homology modeling can be found below.

### Alignment generation
All sequence alignments were generated using structural alignments in Pymol using the align plugin between rGLUT5 4ybq and the target structure, with an RMSD cutoff of 2 Å. The subsequent sequence

alignment from this was then exported and converted to a MODELLER format.ali file using Jalview. Structural alignment was preferred to sequence alignment for this as it ensures structural motif preservation. It was noticed that gaps were produced in helices when a conventional sequence alignment was used. Additionally, structural alignment was useful when certain pieces of a region needed to be modeled, such as in the TM1b region/TM1–TM2 loop in the outward-open state (see below), or when multiple templates are needed for complete coverage, such as in the occluded, inward-occluded, or inward-open states. All alignments used for modeling can be found in on the project OSF repository under 'homology modeling' at https://doi.org/10.17605/OSF.IO/8SJPB.

### Outward-open state

First, the Fv antibody bound to the crystallographic state was removed. The unresolved C-terminal end of TM1b and TM1–TM2 loop (residues Y33–N60) from chain A of the outward-open crystal structure of rGLUT5 (PDB:4ybq) was modeled using TM1b and TM1–TM2 loop (residues Y26–P53) of the crystal structure of GLUT3 (PDB:4zwc) as a template. The GLUT3 TM1–TM2 loop is of 23.5% identity to the rat GLUT5 TM1–TM2 loop. The sequence alignment contained five additional residues from GLUT3 4zwc (residues P54–L58) to align to the resolved TM2 region of GLUT5 4yb9 (residues I33–T37), to aid in proper continuity of the loop. Two hyndred models were generated using the automodel function from MODELLER. Model #2 was chosen manually for subsequent MD simulation, based on retained integrity of an outward-open TM1b, with a TM1b region matching 4zwc as closely as possible. This model had a DOPE score of −64,314.85547. All 200 homology models had a DOPE score between −64,000 and −65000.

### Outward-occluded state

Human GLUT3, 4zw9 (*Deng et al., 2015*), served as template for the rGLUT5 outward-occluded model, as this is the outward-open structure with the closest sequence identity to GLUT5 (an alternative would have been XylE, which has 24% sequence identity compared to 41%, see *Figure 1—figure supplement 1*; *Figure 1B*). GLUT3 (4zw9) and rGLUT5 (4ybq) were structurally aligned (see above) to produce a sequence alignment. As the 4zw9 structure is of high resolution and all loops/IC helices are present in the crystal structure, only one model was created using the automodel function from MODELLER. This model had a DOPE score of −65,212.92969, and all structural features (salt-bridge residue contact, extracellular/intracellular gate) were as expected.

### Occluded state

*Pf*HT1 PDB:6rw3 (*Qureshi et al., 2020*) chain C served as template for the rGLUT5 fully occluded model, as this was the only available structure for sugar porters in the occluded state. PfHT 6rw3 and rGLUT5 4ybq were structurally aligned (see above) to produce a sequence alignment. ICH5, which is partially missing in the PfHT 6rw3 structure, was modeled using residues from F467–V480 from rGLUT5 4ybq. Residues M457–T466 (rGLUT5, 4ybq) were included in the alignment as well to aid in continuity of the ICH5 from TM12. Ten homology models were generated using the automodel function from MODELLER, all with a DOPE score between −59,000 and −60,000. Model 1 was chosen as the TM9–TM10 loop was furthest from the protein center (to avoid a starting conformation with an unphysical conformation).

### Inward-occluded state

XylE PDB:4ja3 (*Quistgaard et al., 2013*) served as template for the rGLUT5 inward-occluded model as this was the only available structure for sugar porters in the inward-occluded state at the time. There were several missing components to this crystal structure that needed supplementation. XylE 4ja3 and rGLUT5 4yb9 were structurally aligned (see above) to produce a sequence alignment. ICH4 and ICH5 which are missing from the XylE 4ja3 structure, were modeled using rGLUT5 4ybq residues A258–R274 and F467–V480, respectively. Residues D253–K257 and M457–T466 were included in the alignment to aid in the continuity of ICH4 between ICH3 and TM7, as well as for the continuity of ICH5 from TM12, respectively. One hundred homology models were generated using the automodel function from MODELLER, all with a DOPE score between −58,000 and −60,000. Model 89 was chosen based on positioning of ICH5, which was modeled closer to the protein relative to the other models,

although the fully resolved ICH5 position is not confirmed structurally in any inward-facing state of the sugar porters to date.

### Inward-open state

Bovine GLUT5, PDB:4yb9 (*Nomura et al., 2015*), served as template for the rGLUT5 inward-open model. As in other structures, ICH5 is not resolved in 4yb9. bGLUT5 4yb9 and rGLUT5 4ybq were structurally aligned (see above) to produce a sequence alignment. ICH5 was modeled using rGLUT5 4ybq residues F467–V480. Residues M457–T466 (rGLUT5, 4ybq) were included in the alignment as well to aid in continuity of the ICH5 from TM12. 200 homology models were generated using the automodel function from MODELLER, all with a DOPE score between −59,000 and −61,400. Model 1 was chosen, with a dope score of −61,280.28516, again based on ICH5 position, as mentioned above in inward-occluded modeling.

### Modeling with automodel

MODELLER's automodel function was used with all default parameters, using DOPE score as an assess method. The script used to generate the models can be found on GitHub at *McComas, 2023*; https://github.com/semccomas/GLUT5_string.

### Protein simulation details

Each rGLUT5 protein model was placed into a POPC bilayer with ~122 lipids on the top leaflet, and ~124 on the bottom, and solvated in a water box with 150 mM NaCl using CHARMM-GUI (*Jo et al., 2008*). For the inward-open simulations, a varied lipid composition was also tested to probe local dynamics effect of lipids using either a bilayer of 100% POPE or a bilayer of 70% POPE, 20% POPG, and 10% DOPA were used instead of POPC. The total box size before equilibration was 10 × 10 × 11 nm. All parameters of the system were described using CHARMM36m.

Each system underwent energy minimization using steepest descent, followed by system equilibration for a total of 187.5 ps where positional restraints on the protein and POPC lipids were gradually released. Production MD was then run using 2 fs timesteps in GROMACS version 2019.1 (*Abraham et al., 2015*). Temperature was maintained at 303.15 K using Nose–Hoover temperature coupling, using three separate groups for protein, lipid bilayer, and the solvent. Pressure was maintained at 1 bar using the Parrinello-Rahman barostat with semiisotropic coupling, using a time constant of 5 ps and a compressibility of $4.5 \times 10^{-5}$ bar$^{-1}$. Hydrogen bonds were constrained using LINCS (*Hess, 2008*), electrostatic interactions modeled with a 1.2-nm cutoff, while long-range electrostatics were calculated with particle mesh Ewald (PME). A brief summary of each simulation including simulation length can be found in *Table 1*. The GROMACS mdp files, index files, and simulations systems can all be found on the project OSF at https://doi.org/10.17605/OSF.IO/8SJPB.

### Targeted molecular dynamics

It was our experience throughout the project that the string simulations were sensitive to the initial path generated. Attempts to generate an initial pathway for the strings included using adiabatic biased molecular dynamics simulations and steered molecular dynamics with several sets of CVs. These initial simulations yielded conformations that were not fully representative of their target state in dimensions other than the CVs. These resulted in FESs from string simulations that kept a clear memory of the starting configuration. For this reason, TMD was chosen as a protocol for generating an initial pathway. As this protocol uses a protein's entire cartesian coordinates as a CV, it has the benefit of generating an initial string in which at least five points along the string were likely to be close to a local minimum along the transport pathway (as structures had been determined for GLUT5 or related sugar porters in these states).

TMD was performed in a stepwise fashion between states to ensure that the initial string would cover the determined sugar porter conformational space observed thus far. Four main TMD protocols were used: rGLUT5$^{empty}$ Outward open – Inward open, rGLUT5$^{empty}$ Inward open – Outward open, rGLUT5$^{fructose}$ Outward open – Inward open, and rGLUT5$^{fructose}$ Inward open – Outward open.

TMD was performed using GROMACS version 2019.5 patched with PLUMED version 2.5.5 (*Tribello et al., 2014*). Each TMD condition was performed in a stepwise, iterative fashion. The first TMD run of each of the four conditions was performed biasing stepwise, either the outward-open structure

or inward-open homology model toward the respective targets, the outward- or inward-occluded models. All structures and models used, be it as a starting point or as a target for the TMD, are from unequilibrated (not from the aforementioned MD) structures/models, to ensure that the TMD was not generated from a local minima distant from a desired state. The positions of all heavy atoms were biased in a geometric space with incrementally increasing harmonic restraints, initially starting at 0 kJ/mol/nm and increasing to 2500 kJ/mol/nm at step 5000. After 5000 steps, the force applied was squared every 150,000 steps until the heavy atom RMSD of the system was within about 0.05 nm of their target conformation. After this was achieved, the final frame of the TMD run was used to generate the next TMD run's input model for each condition. *Table 2* details each TMD run length, starting and ending conformations for each condition, and the RMSD of the final TMD timepoint.

For rGLUT5$^{fructose}$ TMD runs, beta-D-fructofuranose was placed in the outward-open structure or inward-open model based on the positioning of glucose in hGLUT1 (4pyp)(*Deng et al., 2015*) after structural alignment with the models in PyMol version 2.5.0. The D-fructose-bound outward-open structure or inward-open model were then briefly energy minimized to ensure no sugar and water atoms clashing during simulation. During TMD runs, D-fructose coordinates were left unbiased. The GROMACS topology and mdp files, as well as the target configurations for each TMD run can be found on the project OSF repository at https://doi.org/10.17605/OSF.IO/8SJPB.

## Limitations of inward- to outward-open simulations in regards to salt-bridge distances

Initially, as described above, TMD was also performed with both rGLUT5$^{empty}$ and rGLUT5$^{fructose}$, from inward- to outward open. However, string simulations performed of these conditions did not converge. Upon examination of features of these simulations, we could see the state-dependent salt-bridge residues losing contact in states where they should not (*Figure 4—figure supplement 3D*), and thus we elected to focus further simulations on GLUT5 influx, as discussed in the main text.

## CVs selection

CVs were chosen based on features that were transferable to other sugar porters, and that would separate different functional states. Two CVs were used for this state differentiation, measuring opening of the extracellular and the intracellular gates, respectively (*Figures 1C and 2A*, *Figure 1—figure supplement 1A*). The distance between the COM of the extracellular gating parts of the transmembrane helices TM1 (residues 36–43) and TM7 (residues 295–301) was used to measure opening of the extracellular gate (referred to as extracellular gate distance). The distance between the COM of the intracellular gating parts of the transmembrane helices TM4 (residues 142–151) and TM10 (residues 392–400) was used to measure opening of the intracellular gate (referred to as intracellular gate distance).

For other state-defining features as CVs, the following measurements were used. The TM7b angle $\theta$ was calculated as the angle between two vectors defined by two groups of residues COM:

$$\theta = arccos\left(\frac{\vec{BA} \cdot \vec{BC}}{\left|\vec{BA}\right|\left|\vec{BC}\right|}\right) \tag{1}$$

where $A$ represents the vector of positions of residues 289–291 COM, $B$ the vector of positions of residues 296–298 COM, and $C$ the vector of positions of residues 304–306 COM.

The TM10b RMSD was calculated as the weighted average of each grid's RMSD of the backbone of residues 391–401 after superposition of all backbone atoms onto the outward-open structure 4yb9.

The state-dependent salt-bridge residue distance is calculated as the minimum distance averaged between residue pairs E151–R407 and E400–R158.

## String preparation

For each of the conditions, snapshots corresponding to points lining the string were extracted from the TMD runs (referred to as beads hereafter). 16 beads were chosen in total, five of which correspond to the outward-open, outward-occluded, fully occluded, inward-occluded, and inward- open models, and based on the first or final frames of the TMD runs. The other 11 beads were chosen to cover

uniformly the CV space between states (*Figure 2D* for rGLUT5[fructose], *Figure 2—figure supplement 2B* for rGLUT5[empry]).

## String method with swarms of trajectories

The string simulations with swarms of trajectories were performed as described in *Fleetwood et al., 2020*, with a brief summary as follows. With the exception of beads 0 and 15 (outward- and inward-open models), which were held fixed and therefore not simulated in each run, each bead along the string underwent several simulation steps in every iteration of the string simulations. *Step 1: short string reparametrization and CV equilibration*. The CV values were extracted for each bead, and the relevant system was equilibrated with a 10,000 kJ nm$^{-2}$ harmonic force potential acting on each CV for 30 ps. *Step 2: swarms of trajectories*. From each bead, 32 swarms were launched in parallel and run for 10 ps each. *Step 3: calculate CV drift for next iteration*. The drift per bead was calculated by measuring the average of the CV distance over the simulation swarm. *Step 4*: Using the updated CV coordinates, the string was reparametrized so that the beads were equidistantly placed along the string, therefore stopping each bead from falling into nearby energy minima. Details of this reparameterization can be found in *Jia et al., 2020*. Then, the iteration was complete and the next iteration could begin, with the initial simulation restraining the system in the reparametrized CV space. The rGLUT5[fructose] simulations were run for a total of 552 iterations, and the rGLUT5[empty] simulations were run for a total of 745 iterations.

The code for running the string simulations with the conditions above, as well as a tutorial and simple system setup and analysis code can be found on GitHub at https://github.com/delemottelab/string-method-swarms-trajectories, (copy archived at swh:1:rev:8a3faf1d0595b-c9322d04802bff2812702e7f59a; *Conesa, 2022*). All simulation parameters of the string simulations were the same as mentioned above, with the exception of GROMACS version (2020.5 instead of 2019.1), and the use of a V-rescale thermostat instead of Nose–Hoover. The GROMACS topology and mdp files, as well as the conformations of each of the 16 beads from the initial string can be found under 'Running string simulations' on the OSF repository at https://doi.org/10.17605/OSF.IO/8SJPB.

## Free energy landscape calculation

The free energy landscapes as depicted in *Figure 2E, F* were calculated from the transition matrix of the swarm simulations in the CV space once they were determined to be in equilibrium (*Figure 2—figure supplement 2C*), after about 100 iterations. Therefore, 452 iterations of data were used to calculate rGLUT5[fructose] FESs, and 645 iterations of data for rGLUT5[empty].

## Notebook associated with free energy landscape calculation

The python notebook, developed by Sergio Perez Conesa and adapted by Sarah E. McComas, which reads trajectory files, calculates time-lagged independent component analysis (TICA) components, performs *k*-means clustering, Markov state model (MSM) calculation, and subsequent KDE projection of the MSM onto the CV's, can be found in a Jupyter notebook on GitHub at https://github.com/semc-comas/GLUT5_string/blob/master/string/analysis/scripts/string_MSM_analysis_TICs_deep_time-dev.ipynb. Below is a summary of the individual components of these free energy landscapes calculations.

## Time-lagged independent component analysis

The data input to the TICA was for all swarms, all beads, iterations 100–745 or 552 depending on if the rGLUT5[empty] or rGLUT5[fructose] energy landscape was calculated. A lagtime of 10 ps was used for the estimator. *Figure 2—figure supplement 7* depicts the distribution of the raw CV distances (panels A and B) and the subsequent TICA-transformed CVs (panels C and D), both colored by bead for clarity.

## *K*-means clustering on TICA values

First, an appropriate number of cluster centers is decided empirically with the following steps. A range of cluster centers of 5, 10, 30, 50, 75, 100, 200, and 500 were tested. The TICA data are clustered using *k*-means clustering, with the number of cluster centers set to the tested number (e.g., 75), using a data stride of 10, 50 maximum iterations of *k*-means clustering, and a uniform initialization strategy. Then, a maximum likelihood MSM estimator is constructed with a lagtime of 1, assumed reversibility, and no stationary distribution constraint. The transitions between cluster centers are calculated using

a transition count estimator with a lagtime of 1 and a sampling count mode (meaning that a transition is counted every stride of lagtime $\tau$). The MSM is then applied to these transition counts. This MSM is then scored using a VAMP (Variational Approach for Markov Processes) score, wherein the MSM is divided into equal size testing and training data, and the error between this cross-validation is reported. This entire process is repeated five times to get an error estimation on the entire VAMP score calculation. The resulting VAMP score and associated error are shown in *Figure 2—figure supplement 7E, F*.

The minimum number of cluster centers yielding a saturated VAMP score for both rGLUT5$^{empty}$ and rGLUT5$^{fructose}$ was 100. The TICA data were then clustered as described above, using *k*-means clustering with a uniform initialization strategy, this time with 100 cluster centers, a stride of 1, and 500 maximum iterations.

## MSM construction

The MSM was constructed as described above, counting the transitions between the 100 cluster centers with a transition count estimator, and applying the maximum likelihood MSM estimator to these transitions. Probability weights for each data point of the trajectory are then calculated, normalized to 1.

## Projection of the MSM

The CVs used to build the MSM (intracellular gate distance and extracellular gate distance) were also used for the projection of the energy surface. The probability density function was estimated using a Gaussian kernel density estimator on the CV distribution, using the weights calculated from the MSM, and a bandwidth of 0.05. The negative log of the probability density function was taken to project the resulting FES in kT on a 55 × 55 grid, using the minimum and maximum values of each CV throughout the simulation as the extent.

## Blocking and bootstrapping for FES error calculation

As simulation data points are not independently measured, statistical error was calculated on the FES to identify the effect of the correlation between the data using blocking on the trajectory frames and bootstrapping of the blocks. First, a number of blocks which yields the highest error was determined; *n* blocks 2, 4, 8, 16, 32, and 64 were tested. The simulation was split into n blocks, of which each block was randomly bootstrapped 200 times. For each bootstrap, the MSM was calculated as described above, using the same 100 cluster centers as previously determined. The FES of this was again determined using a KDE on the bootstrapped data points and the weights from the MSM with the same parameters as previously described. The error was calculated by measuring the difference between the highest density intervals spanning 95% of the distribution, divided by their mean. The average FES error for each number of blocks rGLUT5$^{empty}$ and rGLUT5$^{fructose}$ simulations can be found in *Figure 2—figure supplement 8A, B*.

From this, it was determined that four blocks yielded the highest FES error. The FES error was then projected as seen in *Figure 2—figure supplement 8C, D*, depicting a maximum error of 3 kT so as to show the errors most meaningful to the FES projection.

## Validation of string simulations

To assess whether the string simulations generate energetically favorable conformations of GLUT5 that are representative of states visited along the transport pathway, several measures were taken. The five homology models are near local energy minima on the 2D projection of the FES (*Figure 2E, F*), but to ensure that other areas of the protein had not deviated dramatically from their initial state, we aligned the homology models to the average conformation in several energy wells (*Figure 2—figure supplement 9*). In this, we could see that the homology models aligned well to the average structure extracted from the energy well.

We then measured the distribution of several state-defining features within these energy wells: TM7b angle, TM10b RMSD, and salt-bridge distance (*Figure 2—figure supplement 10*). We could see here that each energy well was distinct from each other, the distribution within the well was not overly broad, and that the distribution of each feature was close to the measurement for the homology model of the representative state.

**Table 4.** Details of conventional simulations from points along the free energy surface.

**Outward-facing controls**

| Replica | Iteration (out of 552) | Bead (out of 16) | Swarm (out of 32) | Simulation length (ns) |
|---|---|---|---|---|
| 1 | 552 | 1 | 2 | 299 |
| 2 | 552 | 1 | 29 | 296 |
| 3 | 552 | 2 | 1 | 311 |

**Inward-facing controls – upper well**

| Replica | Iteration (out of 552) | Bead (out of 16) | Swarm (out of 32) | Simulation length (ns) |
|---|---|---|---|---|
| 1 | 552 | 10 | 6 | 384 |
| 2 | 552 | 10 | 19 | 383 |
| 3 | 552 | 10 | 23 | 380 |

**Inward-facing controls – lower well**

| Replica | Iteration (out of 552) | Bead (out of 16) | Swarm (out of 32) | Simulation length (ns) |
|---|---|---|---|---|
| 1 | 300 | 11 | 12 | 349 |
| 2 | 551 | 11 | 16 | 348 |
| 3 | 551 | 11 | 24 | 350 |

## Conventional MD simulations from FES conformations

Controls of the string simulations were performed in order to validate the compatibility of the conformations generated from the energy surfaces. Configurations for rGLUT5$^{fructose}$ were randomly taken from regions with low free energy, using the aforementioned FES bins as a conformational ensemble to sample from. Later iterations from the string simulations were preferred for control simulations as these should be the most equilibrated (alternatively, the most affected by hysteresis if present). Conventional MD simulations were performed as described above, preserving the velocities from the swarm simulation for that particular conformation, fructose was maintained in these simulations. No equilibration MD was run for these configurations. *Table 4* details the simulation starting configurations and total simulation length for each control. The starting configurations for these simulations can be found in the OSF repository under 'Analyzing string simulations/String controls starting points', https://doi.org/10.17605/OSF.IO/8SJPB. The distribution of these control simulations from each along the gate CV space can be found in *Figure 2—figure supplement 3* (nine controls in total).

## Reweighting and projection of the free energy landscape on other CVs

FESs were also projected onto other CVs such as TM7b angle, TM10b RMSD, or average salt bridge (*Figure 2—figure supplement 4*). These features were calculated as described in CV selection above. The MSM was also calculated as described above, with one exception: the TM10b RMSD was not found to be continuous across the transport pathway in rGLUT5$^{fructose}$, and thus there was no overlap between the outward-facing states (beads 1–7) to the inward-facing states (beads 8–16) in TICA space. The MSM is still calculable, but discards the less populated clusters which are not connected to the more populated clusters (the outward-facing states) (*Figure 2—figure supplement 4B*). To overcome this, the MSM was calculated for beads 1–7 and 8–16 independently, and their resulting energy surfaces were projected on the same image (*Figure 2—figure supplement 4C*). This makes the two regions calculated independently disconnected and their relative free energies can thus not be determined. To measure the stability of the conformations generated from the string simulations, as well as to validate the free energy wells in alternate CV spaces, we projected the movement of the nine string control simulations onto the different FESs (*Figure 2—figure supplement 3*, *Figure 2—figure supplement 5*, and *Figure 2—figure supplement 6*).

## Analysis of simulations

The resulting free energy landscape is defined on a 55 × 55 grid. For the analysis of protein features (such as sugar coordination, TM7b angle, and salt-bridge distances), each of the bins in the grid

was analyzed independently. For this, structural snapshots from the endpoints of swarm simulations corresponding to the CV values of each bin were extracted, with a maximum of 1000 frames per bin (*Figure 2—figure supplement 11A*).

In the sugar coordination analysis, the snapshots extracted for each bin were aligned on the entire protein position in cartesian space. Then, clustering was performed on each of the sugar coordinates using gmx cluster with the Jarvis–Patrick algorithm and a cutoff of 0.08 nm for each cluster center. The percentage of total frames occupied by the most populated cluster is presented in *Figure 3A*. *Table 3* summarizes the clusters for two representative bins, the outward-open and occluded states. The area shaded gray in the table indicate clusters used in the *Figure 3A* inserts, highlighting the most populated clusters summing to ~70% of the total possible sugar poses in a bin. The most populated cluster for the occluded state, highlighted with a red border, is shown in *Figure 3C*. Distances between certain sugar hydroxyl groups and residue side chains, as shown in *Figure 3D*, are calculated from this cluster as well. These measurements are the minimum closest distance between any atom of a given hydroxyl group, to any atom of a residue side chain for each frame in this bin ($n$ = 447).

In other feature analysis such as TM7b angle or salt-bridge distances, the snapshots extracted for each bin were analyzed in Python version 3.8.5 using MD Analysis version 2.0.0 (*Pao et al., 1998*) and plotted in matplotlib version 3.3.4.

Images overlaying simulation features with an energy surface (such as *Figures 3A and 4C*) use an abstraction of the FESs as depicted in *Figure 2E, F*. A depiction of this abstraction can be found in *Figure 2—figure supplement 11B*.

To estimate average properties for each grid point ($X_i$), weighted averages ($W$) are reported, using the weights for each snapshot estimated from the MSM ($w_i$).

$$W = \frac{\sum_{i=1}^{n} w_i X_i}{\sum_{i=1}^{n} w_i} \tag{2}$$

TM7b angle, TM10b RMSD, and intracellular salt-bridge distance were calculated as defined above in describing CV selection.

The water density as seen in *Figure 3B* was generated from a trajectory of snapshots from the most occupied sugar pose cluster of the occluded state (red outline, *Table 3*). The density was calculated in VMD using the VolMap tool, measuring average water density in all frames, of waters within 5 Å of the D-fructose molecule. The resulting density is visualized in PyMol, with an isomesh density cutoff of 0.7.

The aforementioned Python codes written for all analysis and free energy landscape calculations can be found at: https://github.com/semccomas/GLUT5_string, (*McComas, 2023*).

## rGLUT5 homolog detection and sequence logo generation

The rGLUT5 sequence was searched using NCBI pBLAST against a non-redundant protein database, with 1000 maximum target sequences and *E*-value threshold of 0.05. The sequences were then aligned with ClustalOmega (https://www.ebi.ac.uk/Tools/msa/clustalo/). From this alignment, a sequence logo was generated with Weblogo3 (https://weblogo.threeplusone.com) locally.

## Functional activity of rGLUT5 mutants

### Construct design and cloning

The full-length sequence of rGLUT5 (Uniprot: P43427) was used for functional assays, with two alterations to the sequence. The deglycosylation mutation N50Y is present, and several C-terminal residues are retained after TEV cleavage. Both are underlined in the following sequence. rGLUT5 WT: AMEKEDQEKTGKLTLVLALATFLAAFGSSFQYGYNVAAVNSPSEFMQQFY<u>Y</u>DTYYDRNKENIESFTLTLLW SLTVSMFPFGGFIGSLMVGFLVNNLGRKGALLFNNIFSILPAILMGCSKIAKSFEIIIASRLLVGICAGISSNVVP MYLGELAPKNLRGALGVVPQLFITVGILVAQLFGLRSVLASEEGWPILLGLTGVPAGLQLLLLPFFPESPRY LLIQKKNESAAEKALQTLRGWKDVDMEMEEIRKEDEAEKAAGFISVWKLFRMQSLRWQLISTIVLMAG QQLSGVNAIYYYADQIYLSAGVKSNDVQYVTAGTGAVNVFMTMVTVFVVELWGRRNLLLIGFSTCLTACIVL TVALALQNTISWMPYVSIVCVIVYVIGHAVGPSPIPALFITEIFLQSSRPSAYMIGGSVHWLSNFIVGLIFPFIQV GLGPYSFIIFAIICLLTTIYIFMVVPETKGRTFVEINQIFAKKNKVSDVYPEKEEKELNDLPPATRE<u>QENLYFQ</u>.

## GLUT5 overexpression and membrane isolation

The sequence (WT or with desired mutations) was cloned into the GAL-inducible vector pDDGFP2, containing a His$_8$ sequence and TEV cleavage site (*Drew et al., 2008*; *Newstead et al., 2007*). Mutants were generated by overlap PCR. The vector was cloned into the *Saccharomyces cerevisiae* strain FGY217. rGLUT5 was overexpressed and subsequently purified as previously described (*Drew et al., 2008*; *Nomura et al., 2015*). Briefly, 12 l of *S. cerevisiae* cells were grown in -URA medium containing 0.1% (wt/vol) D-glucose. Cells were incubated at 30°C, inducing with 2% (wt/vol, final) D-galactose once OD$_{600}$ reached 0.6. Twenty-four hours after induction, cells were harvested, resuspended in buffer containing 50 mM Tris–HCl (pH 7.6), 1 mM ethylenediaminetetraacetic acid (EDTA), and 600 mM sorbitol, and lysed mechanically. Unlysed cells and debris were removed by centrifugation at 4°C and 10,000 × *g* for 10 min, then membranes were then isolated by ultracentrifugation at 4°C and 195,000 × *g* for 2 hr. Isolated membranes were then homogenized in 20 mM Tris–HCl (pH 7.6), 300 mM sucrose, and 0.1 mM CaCl$_2$.

## Rat GLUT5 purification

Membranes were solubilized using 1% *n*-dodecyl-β-D-maltopyranoside (DDM, Glycon) in equilibration buffer containing 1× phosphate-buffered saline, 150 mM NaCl, and 10% (wt/vol) glycerol for 1 hr. Unsolubilized membranes were removed by ultracentrifugation at 4°C and 195,000 × *g* for 45 min. The remaining supernatant was incubated with 20 mM imidazole and 15 ml Ni-NTA resin (QIAGEN) for 3 hr with mild agitation. This solution was then transferred to a 30-ml gravity-flow column (Bio-Rad). Immobilized protein was then washed twice with a solution containing equilibration buffer, 0.1% DDM, and increasing concentrations of imidazole: 20 mM then 30 mM. 30 ml of protein was eluted with equilibration buffer containing 250 mM imidazole. The eluted sample was dialyzed overnight in a buffer of 20 mM Tris–HCl (pH 7.6), 150 mM NaCl, and 0.03% DDM, adding equimolar ratio of TEV to the dialysis bag. Dialyzed sample was passed through 5 ml HisTrap columns (GE Healthcare) to isolate cleaved rGLUT5 protein, which was then concentrated (50 kDa MWCO, Amicon) and injected to a Superose 6 column using a flow rate of 0.4 ml/min in a buffer of 20 mM Tris–HCl (pH 7.6), 150 mM NaCl, and 0.03% DDM. Fractions corresponding to the monomer peak of GLUT5 were pooled (*Figure 4—figure supplement 1D*) and concentrated to 2 mg/ml before flash freezing aliquots of protein into liquid nitrogen.

## GLUT5-proteoliposome transport assay

rGLUT5 protein was incorporated into proteoliposomes and radiolabeled sugar uptake measured as recently described (*Saudea et al., 2022*). Briefly, liposomes were prepared in 500 μl batches by mixing total bovine brain lipid extract (Sigma-Aldrich, final 30 mg/ml) and cholesteryl-hemisuccinate powder (Sigma-Aldrich, final 6 mg/ml) were mixed in a buffer of 10 mM Tris–HCl (pH 7.6) and 2 mM MgSO$_4$. The lipid mix was then subjected to several rounds of flash-freeze and thawing cycles, and then sonication. Large lipid particles were removed by centrifugation at room temperature and 16,000 × *g* for 15 min. 20 μg of purified rGLUT5 protein was added to 500 μl of liposomes, and was flash-frozen, then thawed, and extruded through a 400 nm filter (LiposoFast, Avestin).

For steady-state and time course measurement 20 μl of prepared proteoliposomes were added to 2 μl of [$^{14}$C]D-fructose (6 μM, American Radiolabelled Chemicals) and incubated for 2 min (time course measurement in 30-s intervals) at room temperature before stopping the reaction with 1 ml of Tris–MgSO$_4$ buffer, and filtering through a 0.22-μm filter (Millipore), washing with 6 ml Tris–MgSO$_4$ buffer. Filters were transferred to scintillation vials, applying 5 ml of scintillation liquid (Ultima Gold, Perkin Elmer) before measuring by scintillation counting. All reported measurements are repeated in triplicates, using empty liposomes as baseline noise, and WT activity at 100% activity.

For measuring glucose transport activity of rGLUT5 mutants, the same procedure as above is performed, using instead 2 μl of [$^{14}$C]-D-glucose (6 μM, American Radiolabelled Chemicals). As above, empty liposomes are used as baseline noise in the experiments, and WT activity for D-fructose transport is used 100% activity.

For the kinetic analysis of Y296F, 20 μl of prepared proteoliposomes were diluted in 20 μl of increasing concentrations of D-fructose (1–30 mM, in Tris–MgSO$_4$ buffer) with constant stochiometric amounts of [$^{14}$C]D-fructose. The $K_M$ and $V_{max}$ values for D-fructose transport could be determined by measuring initial transport velocities at 60 s. The recorded decay counts of the transported D-fructose

were baseline corrected with counts from empty liposomes and fitted to Michaelis–Menten kinetics using nonlinear regression by GraphPad Prism 9.0.

## Acknowledgements

We thank Sergio Perez Conesa for support using the string of swarms method and related discussions. We thank Albert Suades for providing purified protein of rGLUT5 mutant A395W. This work was funded by grants from Novo Nordisk foundation (to DD) and The Knut and Alice Wallenberg Foundation (to DD) and Göran Gustafsson foundation (to DD and LD). LD acknowledges SciLifeLab and the Swedish Research Council (VR 2019-02433) for funding. The MD simulations were performed on resources provided by the Swedish National Infrastructure for Computing (SNIC) on Beskow at the PDC Center for High Performance Computing (PDC-HPC).

## Additional information

### Competing interests

Lucie Delemotte, David Drew: Reviewing editor, eLife. The other authors declare that no competing interests exist.

### Funding

| Funder | Grant reference number | Author |
| --- | --- | --- |
| Knut och Alice Wallenbergs Stiftelse | | David Drew |
| Novo Nordisk | | David Drew |
| Vetenskapsrådet | | Lucie Delemotte |

The funders had no role in study design, data collection, and interpretation, or the decision to submit the work for publication.

### Author contributions

Sarah E McComas, Data curation, Formal analysis, Investigation, Methodology, Writing – original draft, Writing – review and editing; Tom Reichenbach, Data curation; Darko Mitrovic, Claudia Alleva, Marta Bonaccorsi, Formal analysis, Writing – review and editing; Lucie Delemotte, David Drew, Conceptualization, Formal analysis, Supervision, Funding acquisition, Writing – original draft, Project administration, Writing – review and editing

### Author ORCIDs

Sarah E McComas http://orcid.org/0000-0002-6855-9295
Tom Reichenbach http://orcid.org/0000-0001-5156-4592
Lucie Delemotte http://orcid.org/0000-0002-0828-3899
David Drew http://orcid.org/0000-0001-8866-6349

### Decision letter and Author response

Decision letter https://doi.org/10.7554/eLife.84808.sa1
Author response https://doi.org/10.7554/eLife.84808.sa2

## Additional files

### Supplementary files

• Transparent reporting form

### Data availability

All data generated or analyzed during this study are included in the manuscript and supporting file; Source Data files have been provided for Figure 4.

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
