## [Editor Report]

The current manuscript investigates the energy landscape of the mammalian sugar porter GLUT5 using enhanced molecular dynamics simulations and biochemical assays. The approach generates important insights into the mechanism of GLUT5 conformational change, and into mechanistic diversity among the GLUT sugar porters more generally. The overall strategy is convincing, and the findings will be of interest to the transporter and membrane biology communities.

---

## [Decision Letter]

**Decision letter after peer review:**

Thank you for submitting your article "Determinants of sugar-induced influx in the mammalian fructose transporter GLUT5" for consideration by *eLife*. Your article has been reviewed by 3 peer reviewers, including Randy B Stockbridge as the Reviewing Editor and Reviewer #1, and the evaluation has been overseen by Kenton Swartz as the Senior Editor. The following individuals involved in the review of your submission have agreed to reveal their identity: Krishna D Reddy (Reviewer #2).

Essential revisions:

1. To show that the differences between the free energy landscapes in Figures 2E and 2F are meaningful, the authors should report a statistical error analysis on the free energy landscapes, for example by calculating the latter with different subsets of the simulation data and evaluating the standard error.

2. Other than statistical errors, the reviewers are also concerned that there may be systematic errors due to slow protein motions and sidechains rearrangements that do not properly equilibrate during the simulation time scale (generating hysteresis). Hence, additional validation of the reliability of the calculated landscapes must be performed. If systematic errors are very high, the free energy landscape would reflect high energy (unphysical) conformations, which are typically present if the initial structures are generated with targeted molecular dynamics (TMD), as in this work. Therefore, the authors should show that conventional MD simulations are compatible with the free energy landscape. If systematic errors are indeed confirmed by the authors, unfortunately, this might entail redoing /redesigning the free energy landscape calculations. Conversely, if the controls show that systematic errors are not large, it would be sufficient that the authors discuss them, together with the potential pitfalls of using a small number of collective variables (just two were used, while in other applications by different authors, these were over ten).

The reviewers' specific concerns with regard to hysteresis/systematic errors include:

a) The fact that simulations convergence depends on the direction (outward-to-inward vs inward-to-outward; the authors describe limitations of inward open to outward open simulations in regards to salt bridge distances) indeed strongly suggests the presence of significant hysteresis/systematic errors. Namely, the lack of reversible formation and disruption of salt-bridges might have artificially over-stabilized outward-facing states (as landscapes are only for outward-to-inward direction).

b) If the free energy landscape in Figure 2E is correct, the reviewers would expect the green fuzzy area (inward open/inward occluded) should have at least overlapped with the blue area (fully occluded). This may indicate that either the MD simulation is too short, in which case the authors should elongate it until equilibration into the closest free energy minimum, or the free energy landscape is not correct.

c) The fact that the inward open X-ray structure is a high energy conformation in the free energy landscape in Figure 2E (bottom right triangle; X-ray structure is without substrate) is also concerning since one would expect it to be in the energetically feasible region (within 2-3 kcal mol from a main free energy minimum) unless the X-ray structure is not a functional state, which seems unlikely.

3. The authors should provide a more detailed description of the computational analysis and/or associated citations. In addition, the authors should provide further details on homology models and simulations based on the latter: number of models generated, criteria to select the final model, side chain refinement prior to simulation, quality scores, etc.

4. The authors should provide additional justification as to why the MD simulated outward-occluded state deviates so highly from the starting model. Are there AlphaFold models, or simulations from other groups, that agree with this inconsistency? Alternatively, is the 'crystallized' state stable in the presence of monoolein?

5. The D-fructose concentration tested in the assays was far below the Km (6uM compared to 10 mM), further complicated by unknown Km values of the mutants. This complicates the interpretation of the relative activities since the activity will be highly dependent on substrate concentration. The authors should consider this caveat in their discussion of the mutant activity.

6. The authors should provide additional discussion or clarification on several points related to the experimental results for the mutants.

a. While the 'quadruple' mutant result is impressive and informative, it is difficult to infer firm conclusions from the data and model presented. It might be expected that this mutation should convert GLUT5 into a glucose transporter – yet this mutant is unable to transport glucose.

b. Could the elevated rate for Y296F (and the quadruple mutant) simply be more efficient salt-bridge breaking?

c. If the authors have tested mutation of the YY motif to SN in GLUT5, it would be informative and interesting to know the result regardless of the outcome, in order to understand whether specificity is encoded by this motif alone, or by more complex global interactions.

d. The importance of N293 and its large conformational change was mentioned a few times. Could the authors include a supplementary figure showing how this residue in particular moves to engage with the substrate in the occluded state? Is there anything to the bimodal distribution of distances to the substrate seen in Figure 3D?

e. There is clearly some activity in the N293A mutation, suggesting that this residue (and thus the interaction with the sugar) is not 'required' for transport. In fact, the MD simulations would support this, as the broken helix state can be visited in the absence of substrate. Thus, the authors should rephrase this point.

*Reviewer #2 (Recommendations for the authors):*

1. I would like to see some additional justification as to why the MD simulated outward-occluded state deviates so highly from the starting model. Are there AlphaFold models, or simulations from other groups, that agree with this inconsistency? Alternatively, is the 'crystallized' state stable in the presence of monoolein?

2. The authors nicely demonstrate the coordination of the substrate binding site, and that a tightly conserved N residue in TM7b (that coordinates with the substrate sugar) appears to induce the broken helix conformation and thus the occluded state. Though these conclusions are generally supported by the experimental data, I have some concerns about the conclusions:

– There is clearly some activity in the N293A mutation, suggesting that this residue (and thus the interaction with the sugar) is not 'required' for transport. In fact, the MD simulations would support this, as the broken helix state can be visited in the absence of substrate. Thus, the authors should rephrase this point.

– The D-fructose concentration tested in the assays was far below the Km (6uM compared to 10 mM), further complicated by unknown Km values of the mutants. Thus, I do not think it is appropriate to overly interpret relative activities from initial rates and would suggest the authors use more careful language when referring to these results.

– While the 'quadruple' mutant result is impressive and informative, I find it difficult to infer firm conclusions from the data and model presented. It almost seems as if the objective was to convert GLUT5 into a glucose transporter – yet this mutant is unable to transport glucose. Furthermore, could the reason for elevated transport in Y296F (and the quadruple mutant) simply be more efficient salt-bridge breaking?

– I assume the authors have tested the mutation of the YY motif to SN in GLUT5. If this mutation was tested, it would be informative and interesting to know the result regardless of the outcome. Is specificity encoded by this motif alone, or are more complex global interactions required for promiscuity?

*Reviewer #3 (Recommendations for the authors):*

Currently reported simulation data do not conform to quality standards for publication. In particular, the authors should report an error analysis on the free energy landscapes, for example by calculating the latter with different subsets of the simulation data and evaluating the standard error. Provided the free energy landscapes are converged, free energy minima should correspond to locally stable conformations in conventional MD simulations, hence additional validation of the reliability of the calculated landscapes must be performed also using the latter technique. A good example that the authors could follow and cite, with appropriate quality standards, is Lev et al., PNAS 2017, E4158-E4167, which is also based on the string method with swarms of trajectories and in which the free energies were also obtained from the transition matrix.

The authors reported that they also obtained the free energy in a similar manner but they neither described in detail the methodology nor they cited any reference in this regard, therefore the results are currently not reproducible. A detailed description thereof and/or associated citations must be provided.

The authors should provide further details on homology models and simulations based on the latter: number of models generated, criteria to select the final model, side chain refinement prior to simulation, quality scores, etc.

The authors mentioned that to model efflux was more difficult because of salt bridge formation. A possibility to solve this issue is to increase the number of variables used in the string method, including for example a variable describing salt-bridges formation and disruption. In the previously mentioned reference for example the authors used tens of collective variables rather than just two as in this work. In this regard, the authors should at least discuss the potential issues of a low-dimensional representation.

Assuming the free energy landscapes are reasonably well converged, a suggestion to improve the impact of this work is to obtain the same landscapes with bound glucose, so as to assess how GLUT5 is less selective towards the latter than fructose. Namely, if the alternating access transition in the empty transporter is rate limiting, occlusion with glucose ought to be significantly destabilized with respect to fructose and in this manner, the authors could effectively demonstrate that substrate occlusion controls selectivity.

[Editors' note: further revisions were suggested prior to acceptance, as described below.]

Thank you for resubmitting your work entitled "Determinants of sugar-induced influx in the mammalian fructose transporter GLUT5" for further consideration by *eLife*. Your revised article has been evaluated by Kenton Swartz (Senior Editor) and a Reviewing Editor.

The manuscript has been improved but there is one minor remaining issue that needs to be addressed, as outlined below:

1) The reply to the reviewers contains an important discussion of the observed deviations from free energy minima and a plausible justification of their possible meaning (page 4, under the heading "Compatibility of free energy surfaces with control simulations"). Although these controls are summarized in the methods of the updated manuscript, it is also desirable that the manuscript text be expanded upon to include the following information, which is currently reported only in the response to the reviewers: (1) a description of the drifts observed during conventional MD simulations and (2) the justification described in the response to reviewers. (Namely, that those drifts may reflect the fact that the collective variables used for the free energy landscapes might be very sensitive to small structural fluctuations.)

*Reviewer #3 (Recommendations for the authors):*

The revised manuscript entails significant improvements over the previous version. Namely, statistical errors have now been calculated and reported in the manuscript. Methodological procedures are now better described and the authors clearly exposed the possible limitations of the simulations results.

The authors have clarified most of the critical points in their reply. One remaining issue is that the new controls reported do not rule out the presence of systematic errors in the free energy landscape. In particular, some of those show that unbiased simulations drift out from free energy minima, while ideally they should show equilibration in a local minima across all relevant collective variables.

This notwithstanding, I understand the complexity of modeling conformational transitions in membrane transporters and the stochastic nature of short conventional simulation (which the authors have clearly outlined).

Despite the possible presence of systematic errors, the controls and validations done by the authors seem to indicate that the difference between the free energy landscapes of empty versus bound transporter is meaningful, which together with the structural interpretation and biochemical experiments is the most important result of this work. Therefore, in my opinion, the reported results are relevant and may provide novel important insights into the mechanism of mammalian fructose transporters.

One last suggestion for the authors is to briefly mention and discuss previous computational work where it has been shown that transport is facilitated by energetic stabilization of an occluded state (e.g. upon substrate binding) (Selvam et al. on a SWEET transporter ACS Cent. Sci. 2019, 5, 1085−1096).

---

## [Author Response]

Essential revisions:1. To show that the differences between the free energy landscapes in Figures 2E and 2F are meaningful, the authors should report a statistical error analysis on the free energy landscapes, for example by calculating the latter with different subsets of the simulation data and evaluating the standard error.

We have calculated the statistical error of the MSM calculation of the simulations with this method, which can be found in Figure 2 —figure supplement 4, and included the means of performing this calculation in the Methods section.

2. Other than statistical errors, the reviewers are also concerned that there may be systematic errors due to slow protein motions and sidechains rearrangements that do not properly equilibrate during the simulation time scale (generating hysteresis). Hence, additional validation of the reliability of the calculated landscapes must be performed. If systematic errors are very high, the free energy landscape would reflect high energy (unphysical) conformations, which are typically present if the initial structures are generated with targeted molecular dynamics (TMD), as in this work. Therefore, the authors should show that conventional MD simulations are compatible with the free energy landscape. If systematic errors are indeed confirmed by the authors, unfortunately, this might entail redoing /redesigning the free energy landscape calculations. Conversely, if the controls show that systematic errors are not large, it would be sufficient that the authors discuss them, together with the potential pitfalls of using a small number of collective variables (just two were used, while in other applications by different authors, these were over ten).

Ideally, one would run unbiased MD simulations. We initially tried to do this using the string method, but the protein would keep on getting stuck in local minima. After about 1.5 years into the project, we came to the conclusion that biased simulations were needed as the conformational changes required were just too large to sample the different states in the time scales of the unbiased simulations. We initially tried a machine learning-based approach to finding the right CVs to connect different states, unsuccessfully. In the end, we decided to base our CVs on the gating helices as both biochemical and structural data show they should be important. We then combined these targeted MD simulations with string simulations to obtain less biased trajectories.

The choice of CVs inherently represents a dimensionality reduction that means a loss of resolution or accuracy. In this project, we were more concerned that the CVs are good enough to model the conformational ensemble along the transport pathway to answer our question of interest, i.e. the coupling between sugar binding and transport. We are very excited by our results as well as that MD simulations were sensitive enough to study a transporter with a low substrate affinity of 10 mM. The outcome from our MD simulations has more than exceeded our expectations: we can see changes in the free energy landscape when D-fructose is included and we can even follow sugar binding and gating and it provides the first clear evidence that the occluded state is a high-affinity state that is equivalent to the transition state in enzymes. Moreover, it was very satisfying to see that Q287 was not contributing to hydrogen bonding of D-fructose, as this is the only position where the loss of an OH group in D-fructose is tolerated. That is, the only other sugar (in addition to D-fructose) that can compete for radioactive D-fructose uptake is 2,5-dihydromannitol, which is a sugar lacking an OH group in the C2 position^1^. More recently, we have now obtained the structure of PfHT1 in complex with the D-fructose analogue 2,5-dihydromannitol (Author response image 1). Although the PfHT1 sugar binding site is different from GLUT5, we indeed see a very similar binding pose of D-fructose coordination to GLUT5 to our MD simulations. We are in the process of writing up this structure in a separate paper, but also think our computational and biochemical approach is sufficient enough to support our conclusions without it.

**Author response image 1. sa2fig1:** Crystal structure of PfHT bound to 2,5 anhydro-D-mannitol.

In addition to these observations, which independently validate our methodological development, we complement our manuscript with the following technical sections that review the internal consistency of our method. Below we discuss our experience with hysteresis during the project, compatibility of the energy landscapes/ conformations generated from the enhanced sampling with conventional MD, and the use of our collective variables.

Conformation stability from enhanced sampling simulations

The states generated by the energy landscape are stable conformations representing conformations that GLUT5 visits during its transport cycle. To validate this, we took frames from each low energy well and measured state-defining features to assess each well. We expect that a state defining feature for a proper energy well will be distinguishable from other states’ wells, and that the distribution within these wells will be narrow. Figure 2 —figure supplement 7 depicts the probability distribution of such features for conformational ensembles extracted from free energy minima. The TM7b angle, TM10b RMSD, and average salt-bridge distance for well 3 correspond to those of the inward occluded or inward open homology models (dotted lines), and are distinguishable from the more outward-facing states. The distributions are also relatively narrow, indicating that the conformations spanning the free energy well are not degenerate.

Compatibility of energy surfaces with initial simulations

If we consider the original conventional MD simulations performed from the homology models (Figure 2B), projecting snapshots extracted from the final 100ns of each simulation (to account for system equilibration) on top of the rGLUT5^empty^ free energy surface computed from the string of swarms simulations reveals that the simulations sample the energy landscape appropriately (Author response image 2). This is especially important to note for the inward-facing energy wells, as if hysteresis was present as a result of the TMD pushing from outward open to inward open, the free energy surface would likely deviate far from the original MD distribution in CV space. An exception to the compatibility of these simulations with the free energy surfaces is the inward-occluded simulations, which will be discussed in detail later in this response.

**Author response image 2. sa2fig2:** Overlay of initial simulations with the rGLUT5^empty^ energy surface. The last 100ns of each simulation is shown as a small colored dot, 1 dot per nanosecond, the initial conformation of the homology model in the larger colored dot. The energy surface is projected with the same levels as shown in Figure 2E, with the colors changed to inverted grayscale for simplicity.

Compatibility of free energy surfaces with control simulations

Taking inspiration from the reference Lev. et al. suggested by reviewer #3, we launched simulations of GLUT5 with fructose bound from various points along the rGLUT5^fructose^ energy surface to assess the compatibility of these surfaces with conventional MD. One would expect that conventional MD simulations do not drift out of an energy well if the conformation is stable and if the collective variable is sufficient in describing energy minima. As Figure 2 —figure supplement 8B and C show, in the IC/EC gate space, the control simulations drift out of their respective energy wells. If this is due to the problem of hysteresis generating unfavorable or unrealistic inward facing conformations, given that the TMD simulation were initiated in the outward facing state, then this should be specific to the inward-facing states, while outward-facing states should stay stable in their wells. Therefore, three positive controls were launched for the outward-facing states (Figure 2 —figure supplement 8A) with the same protocol. Surprisingly, we also observed that the outwardfacing simulations did not remain in their energy wells. This leads us to believe that – as the reviewers suggested – using only two distances as collective variables has limitations; one being that they can be overly sensitive to miniscule conformational changes.

To confirm this and validate that our free energy landscapes are indeed converged, we reprojected the free energy landscapes onto several other CVs such as TM7b angle, TM10b RMSD, and salt bridge distance, introduced in the main text (calculations for each feature is as described in manuscript methods). The resulting free energy surfaces for both rGLUT5^empty^ and rGLUT5^fructose^ can be found in Figure 2 —figure supplement 9A and B. In the TM7b angle vs TM10b RMSD calculations for rGLUT5^fructose^, there were not enough transitions between the inward and outward facing states to properly calculate the MSM, resulting in a lack of information for the outward facing states (Figure 2 —figure supplement 9C). As we are interested in the behavior of the outward-facing states for the positive controls, we split the simulations into two and calculated MSMs for each group and thereby obtained disjointed free energy landscapes. We then gathered the results together into one graph so that the relative energy minima for the outward-facing or inward-facing states are visible together. We mention this detail to note that in Author response image 4C and later R6, the free energy difference between outward facing (green in Figure 2 —figure supplement 9C) and inward facing (red in Figure 2 —figure supplement 9C) free energy minima are not reliable.

Projecting the control simulation snapshots onto these energy surfaces in these new CV spaces showed that the controls stayed much more within their own well or that of a neighboring well (Figure 2 —figure supplement 10, 11).

From this, we conclude that the conformational ensembles are indeed converged and that the conformations generated during the string simulations are plausible, including those generated via TMD, namely inward-facing states.

Although the extracellular and intracellular gates distance provides a convenient means to visualize the separation between structural states and the progression of the transporter along its functional cycle, they are not optimal collective variables to define metastable states in an energy landscape. This is in part due to the fact that this CV comes from the measurement of a single distance between centers of mass, making them degenerate and highly sensitive to even small fluctuations in protein movement. Notably, these collective variables had been previously used in string simulations of sugar porters^2^, and we do indeed observe that they separate the 5 initial state models well. In our experience, the string simulations are less sensitive to CV choice than other methods, given that the starting path for the strings is close enough to the minimum free energy path. Thanks to MSM calculations, on the other hand, projection of the conformational ensemble onto a different set of CVs is easy and can be informative, as illustrated in Figure 2 —figure supplement 9. For other enhanced sampling methods, however, a set of CVs that more accurately separates metastable states is likely needed (see accompanying paper).

We thank the reviewer for the thorough comments about the validity of the energy landscapes. In the manuscript, we have addressed the limitations of using two collective variables compared to several, and our reasoning for using these in our work. We also elaborate that the targeted MD was specifically chosen to avoid hysteresis and expand on our experience with this, and provided all figures discussed in this section as new supplementary figures to Figure 2.

The reviewers' specific concerns with regard to hysteresis/systematic errors include:a) The fact that simulations convergence depends on the direction (outward-to-inward vs inward-to-outward; the authors describe limitations of inward open to outward open simulations in regards to salt bridge distances) indeed strongly suggests the presence of significant hysteresis/systematic errors. Namely, the lack of reversible formation and disruption of salt-bridges might have artificially over-stabilized outward-facing states (as landscapes are only for outward-to-inward direction).

The stabilization of the outward facing state is in agreement with the function and structure of GLUT transporters. Monitoring sugar efflux is expected to be complicated due to the challenges of modeling salt-bridge formation vs salt-bridge breakage. Furthermore, sugars are thought to bind with 10-lower affinity on the inside^3^, indicating that a large energetic barrier exists in the transition from inward to outward facing (likely due to the formation of these very salt bridges). We therefore do not believe that the challenges with modeling transport directionality has a connection to systematic errors. As discussed in the manuscript, the outward-facing states have been show to be the most populated in the absence of sugar^4-7^, since the salt-bridge network is asymmetric and only found on the cytoplasmic side.

Biological considerations aside, hysteresis was a concern of ours during the course of this project as mentioned above, and after trying many options for generating initial configurations to launch the swarms from, we found the use of our stepwise TMD protocol to be a suitable way to combat this. Earlier attempts to obtain an initial path for the string simulations included resorting to adiabatic MD (ABMD) as well as steered MD (SMD) on several different groups of collective variables performed in a stepwise manner (moving from one of the five major states to the adjacent one). ABMD was unsuccessful at generating an initial pathway, as the target CV was often not reached during the simulations. In the cases where the target was reached, the RMSD to the target state never decreased throughout the simulation. The SMD performed better in reaching target distributions and RMSD than ABMD, so string simulations initiated from the initial pathway generated by SMD were launched. However, these were discarded after observing a strong effect of hysteresis on the energy surfaces (Author response image 3) even after over 200 iterations of string simulations. From this, it became apparent that applying SMD led to not all degrees of freedom necessary to facilitate a proper conformational change being able to relax appropriately over the subsequent simulation timescales.

**Author response image 3. sa2fig3:** The effect of hysteresis on two string simulations with starting paths generated by steered MD. Both have a strong memory of their initial starting states which are predicted to be the most energetically favorable (red), moving only uphill in free energy.

We therefore elected targeted MD to generate an initial path for several reasons. Firstly, little is known about the precise conformation of GLUT5 during its transport cycle, but the five structures/ homology models should be a close approximation of the major states visited in this cycle. As it was clear that hysteresis was an issue when not all degrees of freedom were known and there was not much information available about the path that GLUT5 takes, targeted MD had the advantage that we could steer the simulations between five presumably reasonable template-based models. Furthermore, as string simulations are performed in equilibrium, they should be somewhat forgiving to imperfect starting configurations. Lastly, targeted MD also has the benefit that it operates on all degrees of freedom, which appeared advantageous for a system for which it was not clear which collective variable to use to properly bias the system. The force applied on the TMD was also performed stepwise so as to allow the protein to take the energetically favorable path to its target.

After applying the string of swarms method to configurations obtained using TMD, projecting free energy surfaces onto various CVs, as shown above in Figure 2 —figure supplement 9, shows that the outward-facing states are not over stabilized. In the free energy surface along the TM7b angle vs average salt-bridge distance (Figure 2 —figure supplement 9A) as well as along the TM7b angle vs the TM10b RMSD (Figure 2 —figure supplement 9B and C), when sugar is present, the inward facing states are more energetically favorable than the outward-facing ones. In turn, in absence of sugar, the outward facing states are shown to be more stable. This corroborates our discussion above where using collective variables describing gate opening may not be appropriate to define metastable states. In contrast, if our simulations truly suffered from hysteresis, the overstabilization of the outward facing states should appear consistently with all projection choices.

On the other hand, we agree with the reviewer that hysteresis is at play in some of our simulations, as the direction of pulling in TMD seems to affect the resulting free energy surface. As TMD pulling does not explicitly take into account all possible CVs , the salt bridge distances, for example, are affected by directionality. Author response image 4A and R4B show the energy surface for the rGLUT5^empty^ and rGLUT5^fructose^ string simulations we have obtained after TMD in the inward open → outward open direction. Though the hysteresis effect of overstabilization of initial states is not as obvious as in Author response image 3, the final states of the path (outward facing) seem to not fully have undergone the proper rocker switch motions (note the improper closing of the intracellular gate in both conditions), likely because pulling in this direction does not result in proper closure of the salt-bridge network as reported in the manuscript. We therefore deemed these conformations unreliable and discarded these simulations from further analysis.

**Author response image 4. sa2fig4:** (A) Free energy surfaces from the MSM for rGLUT5^empty^ with a pathway generated from TMD in the direction opposite to the one reported in the manuscript (in open → out open). Iterations 100 – 420 shown here. B. Free energy surfaces from the MSM for rGLUT5^fructose^ with a pathway generated from TMD in the opposite direction as reported in the manuscript (in open → out open). Iterations 100 – 553 shown here.

b) If the free energy landscape in Figure 2E is correct, the reviewers would expect the green fuzzy area (inward open/inward occluded) should have at least overlapped with the blue area (fully occluded). This may indicate that either the MD simulation is too short, in which case the authors should elongate it until equilibration into the closest free energy minimum, or the free energy landscape is not correct.

We have extended the inward occluded simulation by 200ns, as well as performed two additional replica simulations of the inward occluded homology model. For all three replicas, it is interesting to note that the simulations do not fall to an energy minima on the free energy landscape when using gate distances as a CV (Author response image 5). When we project the movement of these simulations onto free energy surfaces for other collective variables, as introduced above, we see that they do generally occupy energy minima (Author response image 5). In the TM7b angle/ salt bridge distance projection (Author response image 5), which separates the inward-facing states particularly well, the simulations fall into an energy well which would be reasonable for an inward-occluded state (Figure 2 —figure supplement 9A), where the salt bridges are on the way to full breakage.

**Author response image 5. sa2fig5:** (A) Movement of three replicas of the inward occluded state homology model on the gate CV space on top of the free energy surface of rGLUT5^fructose^. A pink dot indicates the conformation for the starting model, and blue dots show the conformation for every 1ns. B. As in panel A, but using TM7b angle vs mean salt bridge distance as the CVs. C. As in panel A, but using TM7b angle vs TM10b RMSD as the CVs.

It is noteworthy that this comparison is being done between inward facing simulations of the homology model which do not contain a bound fructose and the free energy surface of fructosebound GLUT5 since the inward-facing conformations for rGLUT5^empty^ do not feature significant energy wells in intracellular or extracellular gate space. It is also worth noting that the error of the energy surface (Figure 2 —figure supplement 4D) is rather high in this region (IC ~1.1, EC 0.8), making conclusions about this region rather uncertain.

As mentioned in the Discussion section in our manuscript, we generally view the inward-facing energy wells and conformations from this paper with skepticism due to the suboptimal lipid composition, which we know plays a crucial role in stabilizing the inward facing states. We anticipate that further work is needed to properly characterize the inward-facing states and subsequent sugar release into the cell.

c) The fact that the inward open X-ray structure is a high energy conformation in the free energy landscape in Figure 2E (bottom right triangle; X-ray structure is without substrate) is also concerning since one would expect it to be in the energetically feasible region (within 2-3 kcal mol from a main free energy minimum) unless the X-ray structure is not a functional state, which seems unlikely.

We expect the inward states to be highly dynamic as they need to spontaneously close, as the saltbridge network is unique to the GLUT family and ultimately defines them, ie., the sugar porter motif^8^. The x-ray structures represent a functional state, yet they are almost always stabilized by lipids or sugars binding to the gating regions. For example, in GLUT1 inward open structures, there are crystallization lipids (PDB: 6THA), or an endofacial inhibitor (PDB: 5EQI) bound to the inner gate, likewise for GLUT4 (PDB:7WSM/ 7WSN). The related bacterial homolog xylose symporter XylE has two known inward open structures (PDB 4JA4 and 4QIQ), both of which lack structural resolution in several intracellular helices. We expect that the right lipids might help to stabilize the inward open state, but it cannot be too stable either. Indeed, our extensive analysis of lipid preferences of GLUT5 has shown its activity is exquisitely sensitive to membrane fluidity^1^.

To provide further computation evidence, four additional replicas of the GLUT5 inward open model have been run, two of which contain a different lipid bilayer, as can be found on Author response image 6. We see that the initial inward open state is not maintained throughout simulation, and tends to relax towards a slightly more closed intracellular gate. We attempted to stabilize the inward-facing state by using two other lipid compositions with a longer equilibration time, to give time for optimal lipid placement. For these simulations, at least on this time scale, we saw no change in tendency to relax the intracellular gate even with a different lipid bilayer. We conclude that there must be a certain lipid binding mode that is not known, or that the placement of the IC helices is not optimal for stabilizing the inward-open state of GLUT5. To characterize the lipid environment through the protein’s transport cycle, while of great interest to us, is not within the scope of this project as exhaustive sampling would be needed.

**Author response image 6. sa2fig6:** . Movement of inward open simulations superimposed on the rGLUT5^fructose^ free energy surface. The model of rGLUT5 in the inward open state was simulated several times to assess the stability of this state in the intracellular/extracellular gate projection. The magenta dot represents the conformation of the model prior to simulation (which does not vary between conditions/replicas), blue dots represent the simulation frames (one per nanosecond), and the yellow dot represents the end frame of the simulation. Simulation time can be found in Table 1.

Although we attempted to communicate this in our initial draft of the manuscript, we thank the reviewer for the comment, and have clarified our belief on the challenges of identifying a “potentially” more stable inward open state in simulation. We have also extended the Methods section to include a comment about our motivation for using targeted MD as an initial pathway generation protocol, and included Figure 2 —figure supplements 6-11 as figures depicting the controls performed on the string simulations.

3. The authors should provide a more detailed description of the computational analysis and/or associated citations. In addition, the authors should provide further details on homology models and simulations based on the latter: number of models generated, criteria to select the final model, side chain refinement prior to simulation, quality scores, etc.

We have now updated the text extending the methods section to add details of homology modeling, MSM construction, FES error calculation, and verification of the string simulation results.

4. The authors should provide additional justification as to why the MD simulated outward-occluded state deviates so highly from the starting model. Are there AlphaFold models, or simulations from other groups, that agree with this inconsistency? Alternatively, is the 'crystallized' state stable in the presence of monoolein?

As we stated in the submission of our manuscript, the outward-occluded GLUT3 (PDB 4zw9) structure is crystallized in the presence of three monoolein molecules, two of which are directly interacting with the extracellular gate^6^. In our previously published work with simulations of this GLUT3 crystal structure^9^ we reported that the structure quickly transitioned to outward-open upon simulation of the monoolein-free structure.

5. The D-fructose concentration tested in the assays was far below the Km (6uM compared to 10 mM), further complicated by unknown Km values of the mutants. This complicates the interpretation of the relative activities since the activity will be highly dependent on substrate concentration. The authors should consider this caveat in their discussion of the mutant activity.

We are measuring passive influx of sugars. The non-energized uptake of sugars into liposomes has proven very difficult to measure robustly. It has taken us close to four years to get GLUT transporter to work well in proteoliposomes and GLUT5 was particularly challenging as its very low affinity for fructose^1^. Unfortunately, we cannot obtain radioactive sugars at mM concentrations and if we dilute our hot sugar with cold sugar to increase the final concentration then we decrease our signal. With a Vmax at close to 20 mM it is difficult to obtain saturation in a time course experiment. For this reason, we measure uptake after 2 mins, rather than the 30s used for glucose (GLUT) transporters with a 10-fold lower Km. Using this approach on PfHT1 (that transports both glucose and fructose) we found that most mutations abolishing glucose uptake (with saturation) were similar to those monitoring fructose uptake (without saturation)^9^. Those mutations showing some activity need to be further assessed in a time-course assay and further with kinetics, wherever possible.

We have included some of this information in our discussion of the mutants.

6. The authors should provide additional discussion or clarification on several points related to the experimental results for the mutants.a. While the 'quadruple' mutant result is impressive and informative, it is difficult to infer firm conclusions from the data and model presented. It might be expected that this mutation should convert GLUT5 into a glucose transporter – yet this mutant is unable to transport glucose.

Yes, as you might expect, the quadruple mutant was created to hopefully shift GLUT5 to transport D-glucose. We are, nonetheless, surprised that it can still work to transport D-fructose. As the transport is overall low in the quadruple mutant, and time course experiments indicate no saturation of fructose transport after 2 minutes incubation (Figure 4 —figure supplement 1), performing kinetic experiments on the quadruple mutant is not possible. Although the full effect of this recovered effect is not fully understood, it does highlight other coupling pathways are possible. We added text to clarify our conclusions from this mutant.

b. Could the elevated rate for Y296F (and the quadruple mutant) simply be more efficient salt-bridge breaking?

That’s an interesting idea. It is possible that the Y296F mutant helps TM7b to close more easily and therefore increases uptake. It is also plausible that the Y296F mutant is better at resetting from an outward-occluded to an outward-open state, which is probably more likely since re-setting is the rate-limiting step in the transport cycle.

c. If the authors have tested mutation of the YY motif to SN in GLUT5, it would be informative and interesting to know the result regardless of the outcome, in order to understand whether specificity is encoded by this motif alone, or by more complex global interactions.

We thank the reviewer for the suggestion, indeed this was an idea we probed early on in our studies of PfHT1, which is more robust to monitor mutagenesis than GLUT5. We found that PfHT1 S315Y (figure 3D in (9)) and PfHT1 N316Y (not published) both had no transport abilities. We also have previously concluded that as rGLUT5 Y287 and Y298 are conserved for all mammalian GLUTs and XylE, which all transport substrate other than fructose, there must be other factors encoding specificity. To finally rule out this possibility, we have purified and tested rGLUT5 Y297S-Y298N in our functional assay. As can now be seen in an updated Figure 4, this mutant yielded no glucose or fructose activity.

d. The importance of N293 and its large conformational change was mentioned a few times. Could the authors include a supplementary figure showing how this residue in particular moves to engage with the substrate in the occluded state? Is there anything to the bimodal distribution of distances to the substrate seen in Figure 3D?

Thank you for the suggestion, this was something that was included in our paper introducing the PfHT1 structure, but would also be beneficial here. This can now be found in Figure 3 supplement 1.

The bimodal distribution in Figure 3D depicts two possible bonding conformations of N293, as there are two major rearrangements of N293 within this clustered sugar position. Either the N293 amino group faces the C4 OH (as seen in Figure 3B), or the carbonyl group does. The relative populations of the arrangements can be seen in Author response image 7.

**Author response image 7. sa2fig7:** The distribution of the distances of the two groups of the asparagine side chains in the clustered sugar position (see Methods).

e. There is clearly some activity in the N293A mutation, suggesting that this residue (and thus the interaction with the sugar) is not 'required' for transport. In fact, the MD simulations would support this, as the broken helix state can be visited in the absence of substrate. Thus, the authors should rephrase this point.

The Asn residue is strictly conserved in all GLUTs and related sugar porters. In papers spanning more than 10 years, almost every residue in GLUT1 was substituted to cystine in a cysteine-less GLUT1 variant showing WT activity. The TM7b asparagine (Asn288 in GLUT1) was found to one of 8 residues that are strictly essential for uptake of 2-deoxyglucose into oocytes^10,11^, i.e., was the most important residue in the sugar binding site. These mutational studies are consistent with both the structure and cold competition assays, since the TM7b Asn residue hydrogen bonds to the most critical C3-OH and C4-OH groups of D-glucose and likely D-fructose^3,9^. However, you are right that we do observe a higher signal than background in GLUT5 proteoliposomes. Our best guess is that the N293A mutant in TM7b might be able to still close during fructose efflux – albeit less effectively. We will explore this further, but we first need to be able to “silence” oppositely orientated GLUT5 in the proteoliposomes to do so. From a physiological context this mutant is nonetheless “dead” as the affinity for D-fructose is likely to be to very low.

Reviewer #3 (Recommendations for the authors):Currently reported simulation data do not conform to quality standards for publication. In particular, the authors should report an error analysis on the free energy landscapes, for example by calculating the latter with different subsets of the simulation data and evaluating the standard error. Provided the free energy landscapes are converged, free energy minima should correspond to locally stable conformations in conventional MD simulations, hence additional validation of the reliability of the calculated landscapes must be performed also using the latter technique. A good example that the authors could follow and cite, with appropriate quality standards, is Lev et al., PNAS 2017, E4158-E4167, which is also based on the string method with swarms of trajectories and in which the free energies were also obtained from the transition matrix.

Thank you for the reference, we have now addressed this in the essential revisions above.

The authors reported that they also obtained the free energy in a similar manner but they neither described in detail the methodology nor they cited any reference in this regard, therefore the results are currently not reproducible. A detailed description thereof and/or associated citations must be provided.

We have now updated the methods and added Figure 2, figure supplement 3 and 4 for showing our methodology in the free energy calculations.

The authors should provide further details on homology models and simulations based on the latter: number of models generated, criteria to select the final model, side chain refinement prior to simulation, quality scores, etc.

The methods have been updated to incorporate this information. On the public repository linked in the Methods section, one can also find the relevant alignment files needed for the homology modeling.

The authors mentioned that to model efflux was more difficult because of salt bridge formation. A possibility to solve this issue is to increase the number of variables used in the string method, including for example a variable describing salt-bridges formation and disruption. In the previously mentioned reference for example the authors used tens of collective variables rather than just two as in this work. In this regard, the authors should at least discuss the potential issues of a low-dimensional representation.

We have addressed this concern in essential revisions.

Assuming the free energy landscapes are reasonably well converged, a suggestion to improve the impact of this work is to obtain the same landscapes with bound glucose, so as to assess how GLUT5 is less selective towards the latter than fructose. Namely, if the alternating access transition in the empty transporter is rate limiting, occlusion with glucose ought to be significantly destabilized with respect to fructose and in this manner, the authors could effectively demonstrate that substrate occlusion controls selectivity.

Thank you for the suggestion. Since D-glucose doesn’t compete for the uptake of D-fructose then it’s unlikely this sugar binds to GLUT5 with any measurable affinity. Given the binding affinity for D-fructose is already very low (10mM), it is unlikely we will be able to obtain any meaningful data for MD simulations with sugar that does not bind. However, this is something we are very interested in for other transporters like PfHT1 that can recognize multiple sugars, but this is beyond the scope of our current work.

[Editors' note: further revisions were suggested prior to acceptance, as described below.]

The manuscript has been improved but there is one minor remaining issue that needs to be addressed, as outlined below:1) The reply to the reviewers contains an important discussion of the observed deviations from free energy minima and a plausible justification of their possible meaning (page 4, under the heading "Compatibility of free energy surfaces with control simulations"). Although these controls are summarized in the methods of the updated manuscript, it is also desirable that the manuscript text be expanded upon to include the following information, which is currently reported only in the response to the reviewers: (1) a description of the drifts observed during conventional MD simulations and (2) the justification described in the response to reviewers. (Namely, that those drifts may reflect the fact that the collective variables used for the free energy landscapes might be very sensitive to small structural fluctuations.)

We have now included text in the results and later discussion describing the drift observed in the simulation and the justification for this drift as mentioned in the review response. We have also re-ordered the methods to follow the order of the manuscript with this inclusion, with some minor text rearrangements. We also re-ordered the figure 2 supplementary figures to accommodate this addition, with no change to the figures themselves.